# Signal and Noise: A Framework for Reducing Uncertainty in Language Model Evaluation

**David Heineman**[μ] **Valentin Hofmann**[μσ] **Ian Magnusson**[μσ] **Yuling Gu**[μ]
**Noah A. Smith**[μσ] **Hannaneh Hajishirzi**[μσ] **Kyle Lo**[μ] **Jesse Dodge**[μ]

[μ]Allen Institute for Artificial Intelligence
[σ]Paul G. Allen School of Computer Science & Engineering, University of Washington
contact: davidh@allenai.org

## Abstract

Developing large language models is expensive and involves making decisions with small experiments, typically by evaluating on large, multi-task evaluation suites. In this work, we analyze specific properties which make a benchmark more reliable for such decisions, and interventions to design higher-quality evaluation benchmarks. We introduce two key metrics that show differences in current benchmarks: `signal`, a benchmark's ability to separate better models from worse models, and `noise`, a benchmark's sensitivity to random variability between training steps. We demonstrate that benchmarks with a better `signal-to-noise` ratio are more reliable when making decisions at small scale, and those with less `noise` have lower scaling law prediction error. These results suggest that improving `signal` or `noise` will lead to more useful benchmarks, so we introduce three interventions designed to directly affect `signal` or `noise`. For example, we propose that switching to a metric that has better `signal` and `noise` (e.g., perplexity rather than accuracy) leads to better reliability and improved scaling law error. We also find that filtering noisy subtasks, to improve an aggregate `signal-to-noise` ratio, leads to more reliable multi-task evaluations. We also find that averaging the output of a model's intermediate checkpoints to reduce `noise` leads to consistent improvements. We conclude by recommending that those creating new benchmarks, or selecting which existing benchmarks to use, aim for high `signal` and low `noise`. We use 30 benchmarks for these experiments, and 375 open-weight language models from 60M to 32B parameters, resulting in a new, publicly available dataset of 900K evaluation benchmark results, totaling 200M instances.

○ allenai/signal-and-noise 🤗 datasets/allenai/signal-and-noise

## 1 Introduction

Language model development is expensive. During the development process, researchers need to make decisions such as what architecture to use, what training methods to employ, and what data to train on. These decisions rely on measuring phenomena at smaller, more economical scales, then hoping the trends measured hold for large scale models. This paradigm exists across the research community; many papers experiment with small baselines then scale up the best-performing model [31, 17, 38, *inter alia*], and there has been extensive research on using scaling laws to predict the performance of larger models [9, 19, *inter alia*]. While there is a large and ever-growing number of benchmarks, prior work has shown these scaling procedures only works for some benchmarks and not others [66, 56, 15, 50]. This poses a significant challenge because, as we develop more general-purpose language models, developers need to be evaluating on even more diverse benchmarks, some of which may not be well-suited for this critical approach. We need a deeper understanding

39th Conference on Neural Information Processing Systems (NeurIPS 2025).

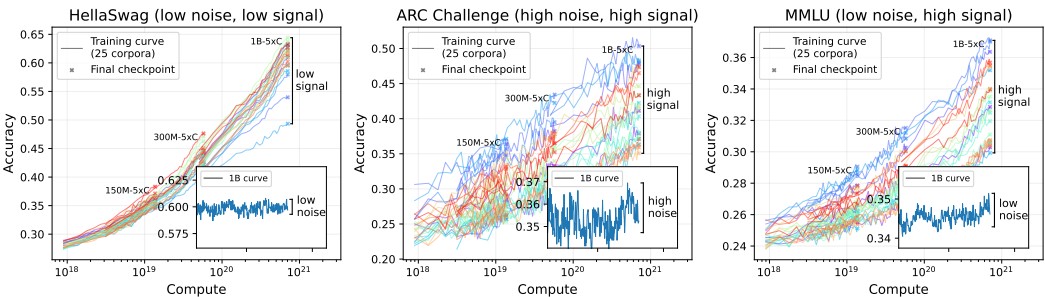

Figure 1: Training curves for the 25 pretraining corpora in DataDecide [38] on three development benchmarks across different model sizes – the ordering of different model pre-training corpora, shown by different colors, at a small scale (e.g., 150M) should agree with ordering at a larger scale (1B), implying better decision accuracy. We hypothesize that one indicator of decision accuracy is the ratio between the `signal` (main plot) and the `noise` of scores within a single training run (inset axis). In this work, we quantify the `signal-to-noise` ratio at different compute scales, and in later sections, show that it is predictive of large scale phenomena like decision-making error.

of what intrinsic properties we can measure to tell if a benchmark provides useful information, if it needs to be reformulated, or if it is best discarded altogether.

To formalize this setup, we study two common experimental settings for language model development: (i) train a pair of small models (e.g., on different pretraining corpora) and use their ranking to predict the ranking of two large models [38], and (ii) fit a scaling law on a set of small models and predict the performance of a large model [19, 3]. We hypothesize that the ability to predict both settings are related to a measure which is cheaper to compute and easier to improve: `signal` and `noise`. Signal measures how spread out scores are for different models on a single benchmark and `noise` measures the variability of a benchmark score during training.

To illustrate the connection from `signal` and `noise` to an experimental setting, consider an example of comparing models trained using different pretraining corpora (illustrated in Figure 1); the tasks where scores are either too close (HellaSwag, left) or too noisy (ARC Challenge, center) are the benchmarks where we would be less confident that a ranking of models at a small scale would hold at a large scale. Following this observation, we show in Section 4 that the `signal-to-noise` ratio (SNR) is highly correlated with the likelihood that a ranking of models at a small scale will hold at a large scale, and then show that `noise` is highly correlated with the prediction error of a scaling law fit.

Based on these observations, in Section 5 we propose a set of interventions designed to reduce `noise` or increase `signal`, and then we measure their impact on our experimental setups of decision accuracy and scaling law error. For example, we show that by averaging out the checkpoint-to-checkpoint `noise` for a model, we improve our ability to predict performance of large models from small models. We also show that it is possible to find subsets of existing benchmarks that have higher `signal-to-noise` ratios than the full evaluation sets, and that even though those subsets can have fewer than half as many instances, they improve both experimental setups. Finally, we show that SNR can be used to improve metric construction, where choosing a metric that has better SNR leads to consistent improvements on a wide variety of benchmarks.

Our core contributions are as follows: (i) we introduce definitions for `signal`, `noise`, and `signal-to-noise` ratio in the setting of evaluating language models, and show this framework is useful for measuring the utility of benchmarks, and (ii) we demonstrate interventions based on this framework which improve both prediction settings. Our core results evaluate 465 language models on 30 benchmarks across 14 model sizes. We release our data, evaluation results, and trained models.

## 2 Predicting Large Model Phenomena with Small Models

Using small scale experiments to make predictions about large model behavior is ubiquitous in language model development [27, 60, 31, 42]. This process can take many forms. For example, finding a good mix of data from multiple sources to train on typically involves evaluation of small models to calculate an optimal weighting of datasets, then training a large model on the optimized

mix [35, 66]. In Blakeney et al. [5], mid-training runs on a sample of candidate pretraining datasets are used to estimate the quality of training from-scratch. Dubey et al. [17] predicted the downstream task using scaling laws to compare candidate data mixes. Hyperparameter transfer methods, such as maximal update parametrization ($\mu P$), also rely on small scale experiments [68]. However, the results from small scale experiments are not always reliable. Work on so-called emergent capabilities [65] shows that for some benchmarks, language model performance only rises above random chance for models trained at large compute budgets. Later work has further explored emergence behavior in particular tasks, such as MCQA tasks [67] or generative math and code tasks [57], or by observing the capabilities of open-weight models [51].

While these different experimental setups are all important, we focus on two straightforward and common setups in making data decisions for language model development: decision accuracy and scaling law prediction error. In this section, we present the motivation for both experimental settings, and in Section 3 we show how the `signal-to-noise` ratio is an effective framework for predicting how useful a benchmark in these scenarios.

## 2.1 Decision Accuracy and Scaling Law Prediction Error

**Decision Accuracy.** Consider a scenario where a practitioner intends to train a large model, and needs to decide between training on Dataset $a$ or Dataset $b$ to get the best performance on some downstream task, represented by a scalar $B(\cdot)$. A simple and intuitive approach is to train a small model $s_a$ on Dataset $a$ and another, $s_b$, on Dataset $b$, then choose the dataset that led to the best downstream task performance for training the large model. We evaluate this procedure by training two large models, $m_a$ and $m_b$, one on each of the datasets, and see if the ranking of the two small models, $s_a$ and $s_b$, on the benchmark is the same as for the large models.[1] In the scenario where we are deciding between more than two choices, we consider pairwise rankings between all pairs $\mathcal{P}$. Following Magnusson et al. [38] we refer to this small-to-large agreement as "decision accuracy":

$$\text{Decision Accuracy} = \frac{1}{|\mathcal{P}|} \sum_{(a,b) \in \mathcal{P}} \mathbb{I}\left[\text{sign}(B(s_a) - B(s_b)) = \text{sign}(B(m_a) - B(m_b))\right] \quad (1)$$

We use models of 7 sizes (from 60M parameters up to 1B parameters) trained on 25 different pretraining corpora from Magnusson et al. [38]. Our prediction task is to use a set of small models (e.g., 60M parameter models) to predict the ranking of the 1B models on a given benchmark (e.g., MMLU). High decision accuracy means the ranking of the small models accurately predicts the ranking of the large models on that benchmark; this is an indication that the benchmark is useful for this process of using small models to make decisions about which dataset to train on. We illustrate an example of this in Figure 1, which shows training curves for 25 data recipes on 3 model sizes. We hypothesize that if model scores are very close together, or the evaluations are very noisy, it is more likely that the ranking from small to large models will change, leading to worse decision accuracy; we formalize and test this hypothesis in the following sections.[2]

**Scaling Law Prediction Error.** Scaling laws [27, 24, inter alia] have been used extensively to predict the validation loss of a large model using a set of smaller "scaling law" models. Recent work has also used scaling laws to predict downstream task performance [19, 3] by first predicting task loss then using the predicted loss to predict task performance (e.g., accuracy); this is the setup we use in this work. The prediction error for the scaling law fit is defined as the relative error between the predicted and true performance of the large model: $\text{Prediction Error} = \frac{|\text{Measured Value} - \text{True Value}|}{|\text{True Value}|}$.

Calculating prediction error requires training a set of scaling law models on the same corpus with varying tokens/sizes (e.g., 190M to 1B params), training a large model (e.g., 13B), and fitting a scaling law to the smaller models to predict the larger model performance.[3] We describe the scaling law functional form and fitting details in App. A.1, following the setup in Bhagia et al. [3].

---

[1]Training multiple large models is too expensive for most development scenarios, but is necessary to evaluate how accurate this process is.

[2]We observe similar findings on other rank agreement metrics, like Spearman rank correlation (Table 3). Decision accuracy, in particular, is equivalent to Kendall's tau modulo a scale and shift (App. A.2).

[3]Scaling law predictions can be used to make development decisions (e.g., about which training dataset is best) by training a set of models and fitting a scaling law for each option being considered [17], but in this work we just evaluate scaling law error directly.

## 2.2 Evaluation Dataset

We perform our analysis using existing development benchmarks and models:

**Models.** Our set of models includes: (i) a suite of scaling law models from 190M to 3.2B, with a corresponding target at 7B and 13B [3], (ii) a suite of 25 models each trained with different pre-training corpora from 60M to 1.3B [38], (iii) the final 30 checkpoints for OLMo 2 1B, 7B, 13B and 32B [42], and (iv) 73 open-weight base models. Additionally, in our comparison between sources of modeling noise in §3.1, we train and release 20 1B models, with 10 models trained varying the data order initialization and 10 varying the random seed initialization, along with evaluation on 3.2K intermediate checkpoints.

**Benchmarks.** We evaluate 30 development tasks which we categorize as knowledge QA, math, and code. We use the OLMES [22] standard where applicable, and reproduce the OLMo 2 evaluation setup [42] for all other benchmarks. Following Gadre et al. [19], we also include multi-task averages for each group, and for the OLMES core tasks. For our test of subset selection in §5.1, we include a synthetically generated benchmark, generated using AutoBencher [32].

We include full details on the sets of models and benchmarks in App. A.5.

# 3 Quantifying `Signal` and `Noise`

To illustrate the impact of `noise` on a decision-making setup, Figure 1 shows training curves for 25 1B models trained with different data recipes and, in inset plots, the training curve for a single 1B model on three tasks. Some tasks (left, HellaSwag) exhibit low `noise` between training checkpoints but low `signal` between models, and others (center, ARC-Challenge) exhibit high `noise` and high `signal`. In this section we define `signal` and `noise`, and define two simple metrics to estimate the `signal-to-noise` ratio that can be calculated from a set of model evaluations on a given benchmark.

## 3.1 Measuring `Noise`

There are numerous sources of noise in the language model development pipeline. Previous work has shown multiple training runs under the same configuration can lead to different performance as a result of a different initialization or data order [14, 13]. In addition, as illustrated in Figure 1, performance can even vary significantly from one checkpoint to the next: within the final 30 checkpoints of training for 1B models on ARC Challenge, we observe a range of 1.7% accuracy. With these motivations, we consider four potential noise measurements, each calculated on using evaluation on a single benchmark: (i) training multiple models and varying only the random initialization, (ii) training models and varying the training data order, (iii) measuring the total checkpoint-to-checkpoint noise across a full, single training run, and (iv) measuring the checkpoint-to-checkpoint noise of the final $n$ checkpoints of a single training run. We formalize these definitions in App. A.3.

To get estimates for four potential sources of noise, we train 10 different 1B-5xC models varying the initialization and data orders, and evaluate all intermediate checkpoints. We find that the initialization noise, data order noise, and checkpoint-to-checkpoint noise across the whole training run all correlate highly with the relative standard deviation of the final $n$ checkpoints ($R^2$ of 0.82, 0.86, and 0.95, respectively, see Figure 7; and see the training curves in Figure 19). These results lead us to define `noise` as the relative standard deviation of the final $n$ checkpoints, as this requires no additional training cost and only uses the final $n$ checkpoints rather than the full training curve. We define `noise` as: Rel. Std.$(m) = \sqrt{\frac{1}{n-1} \sum_{i=1}^{n} (m_i - \bar{m})^2} / \bar{m}$.

## 3.2 Measuring `Signal`

A benchmark is most useful during language model development if it can detect a true difference between a good model and a poor model, assuming a true difference exists between the models in the ability that the benchmark aims to measure. This statistical power is what enables us to use small models for development decisions like training dataset to use. To formalize this idea, we consider a benchmark to have high signal when models evaluated on it have a wide and evenly distributed range of scores. We measure signal using a metric from the numerical integration literature: dispersion, calculated as the maximum difference between the scores of any two models, divided by the mean

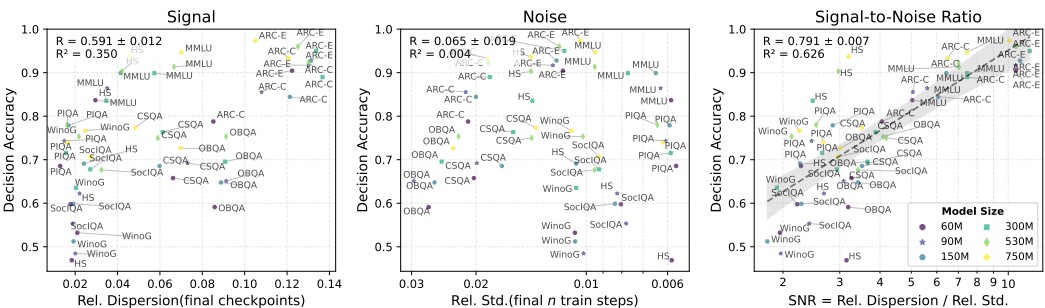

Figure 2: `Signal`, `noise`, and `signal-to-noise` ratio ($x$-axis) vs. decision accuracy ($y$-axis), (see Section 2 for definitions). The `signal` alone (left) and `noise` alone (center) have low correlation with decision accuracy, while the `signal-to-noise` ratio (right) is correlated with decision accuracy. The `signal-to-noise` ratio gives us information about wether a benchmark is useful during development, as high decision accuracy (and `signal-to-noise` ratio) means development decisions made at a small scale generalize to large scale models.

score of all models to account for different scales. This metric is designed specifically to measure how well a set of points cover a space; that is, how spread out the points are from each other. We also considered 20 different measures of spread, including variance, mean pairwise distance, Gini coefficient, etc., in Appendix A.4.

In the following section we introduce `signal-to-noise` ratio, and find that this definition of `signal` leads to `signal-to-noise` ratio with the highest correlation with decision accuracy. We define `signal` as Rel. Dispersion$(M) = \max_{j,k} |m_j - m_k|/\bar{m}$, the normalized maximum difference between any pair of models $j$, $k$.

### 3.3 Measuring `Signal-to-noise` Ratio

Using our measures of `signal` (§3.2) and `noise` (§3.1), we propose measuring the `signal-to-noise` ratio. For both measures, we first divide by the average to be independent of particular units (e.g., to compare accuracy to unbounded task perplexity). We define the `signal-to-noise` ratio:

$$\text{Signal-to-Noise Ratio} = \frac{\text{Rel. Dispersion(final train checkpoint)}}{\text{Rel. Std.(final } n \text{ train checkpoints)}} \tag{2}$$

where `signal` (Rel. Dispersion) is measured over a population of models trained using a similar compute budget, and `noise` (Rel. Std.) is measured over the final $n$ intermediate training checkpoints of a single model. We emphasize that, while this is one particular instantiation of the `signal-to-noise` ratio, our framework is designed to be independent of a particular metric: we find many other measures of `signal` produce similar results in Appendix A.4 and measures of `noise` have high correlation in Appendix A.3.

## 4 `Signal` and `Noise` Correlate with Better Predictions

In this section, we show that the `signal-to-noise` ratio correlates with decision accuracy for small scale experiments, and that the `noise` of the target model correlates with scaling law prediction. These findings motivate our use of SNR to improve benchmarks' statistical properties in Section 5.

### 4.1 Higher `signal-to-noise` ratio indicates higher decision accuracy

**Setup.** We hypothesize that a higher `signal-to-noise` ratio makes it easier to distinguish between models. To test this, we measure decision accuracy using the ranking of the small DataDecide models (60M to 750M) to predict the ranking of the large DataDecide model (1B). To calculate `signal` we use the final checkpoint of each of the 25 small models, and to calculate `noise`, we use the standard deviation around the final 5 checkpoints of the small-scale models. Since we have a measure of `noise` for *each* model, we use the average of the `noise` across the small models.

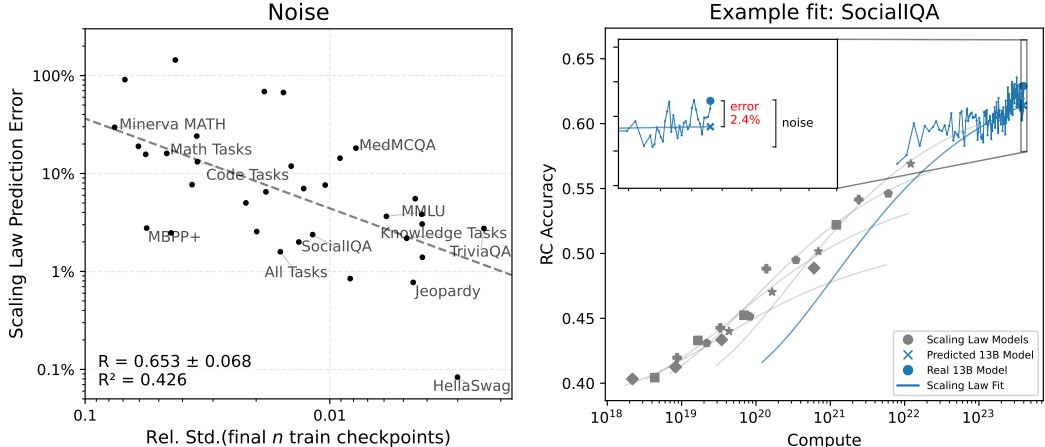

Figure 3: **Left:** Correlation between the `noise` and scaling law prediction error (see Section 2 for definitions). We observe benchmarks with a lower `noise` around the scaling law target ($x$-axis) also exhibit lower error ($y$-axis). **Right:** Example of scaling law for one benchmark (SocialIQA), with examples on all benchmarks in Figure 15. We conjecture that the `noise` of the target model (see inset axis) acts as a bound on the true minimum scaling law error; if the observed scaling law error below this `noise`, then the error is only possible by random chance. Therefore, when benchmarks exhibit a similar scaling law error but different `noise` (e.g., MBPP+, SocialIQA and TriviaQA; see Figure 15), we argue that those with the lowest `noise` are better.

**Signal-to-noise is predictive of decision accuracy.** Figure 2 shows the `signal`, `noise` and `signal`-to-`noise` ratio plotted against the decision accuracy across the OLMES benchmarks. While the `signal` or `noise` alone do not correlate with decision accuracy, we find a strong correlation between SNR and decision accuracy ($R = 0.791$, $R^2 = 0.626$). We conclude that benchmarks which have higher SNR at small scales exhibit higher decision accuracy, and are more likely for their results to hold at a larger scale. In Appendix B.1, we observe benchmarks with a higher SNR also exhibit lower variance when calculating decision accuracy using different checkpoints around the end of training.

## 4.2 Tasks with higher `noise` also have higher scaling law error

**Setup.** We fit scaling laws to predict the performance of OLMo 2 13B using final checkpoint of the set of scaling models trained by Bhagia et al. [3]. We calculate the scaling law prediction error as the relative error of the predicted and final 13B checkpoint. To estimate the noise, we calculate the relative standard deviation of the final 30 checkpoints of the 13B training run, each spaced 1000 training steps until the end of training.[4] We hypothesize that the range of the final $k$ checkpoints of the prediction target (the large, 13B model) acts as an lower-bound on the true minimum scaling law prediction error. An example of the prediction error and noise around the prediction target is illustraed using SocialIQA in Figure 3 (right). Assuming a scaling law with no bias, we expect tasks with a lower standard deviation of the prediction target to also have a lower prediction error.

**Noise measures the reliability of scaling law prediction errors.** In Figure 3 (left), we show the scaling law error and standard deviation for predicting the 13B model performance over 30 tasks. We observe a correlation between the standard deviation of the prediction target and the prediction error across tasks ($R = 0.653$, $R^2 = 0.426$), however the fit is not perfect. For example, we observe four tasks (MBPP+, SocialIQA, MMLU and TriviaQA) which exhibit similar error (around 2–3%), but exhibit different amounts of noise around the prediction target. For these benchmarks with similar error but lower `noise`, we can be confident that the error we observe from the single scaling law fit is the result of the true error of the scaling law fit rather than random chance. In practice, we recommend practitioners prefer making decisions based on scaling law predictions using tasks with low error *and* low `noise`.

Previous work has fit multi-task averages to predict scaling laws. In particular, Gadre et al. [19] find that the error from the individual tasks in their work to be too difficult to predict accurately. In Figure

---

[4]We found 30 checkpoints to be an adequate trade-off between sample size and compute cost. We provide guidance on selecting $n$ when calculating noise, and its impact on experimental results, in Appendix A.3.2.

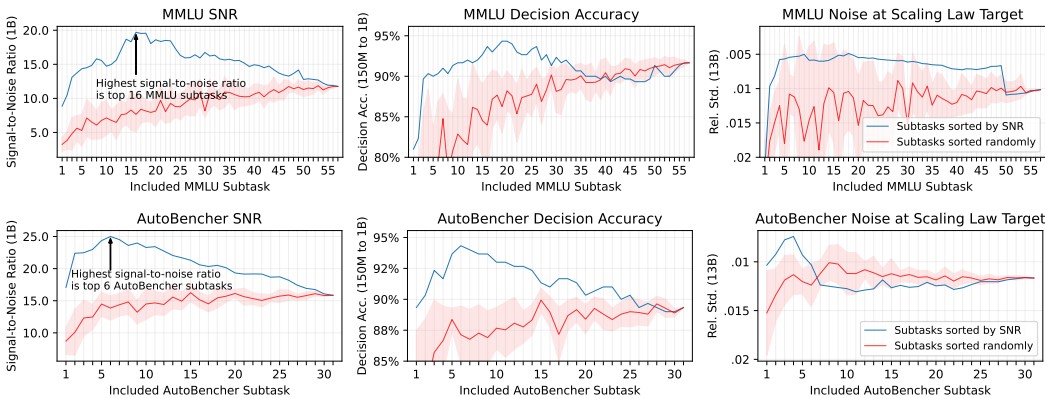

Figure 4: Evaluating an intervention designed to increase `signal-to-noise` ratio (SNR): selecting subsets of a benchmark (Top: MMLU; Bottom: AutoBencher) that have higher SNR dramatically improves decision accuracy and the noise of the scaling law prediction target. MMLU and Auto-Bencher are made of different subtasks; for each benchmark we sort its subtasks by their SNR, then greedily add subtasks to our subset in order of decreasing SNR (left to right). Despite the subsets made in this way having fewer test instances, we find subsets of MMLU (e.g., with 16 subtasks) and of AutoBencher (e.g., with 6 subtasks) that have higher SNR than the full sets, and also have better decision accuracy and noise around the scaling law target. Named subtasks in Figure 16 in Appendix.

3 we also plot results for multi-task averages for each task group ('Knowledge', 'Math', 'Code') and an average across 'All Tasks'. We find that some individual tasks are easier to predict than multi-task averages, and have lower noise around the prediction target. In particular, generative tasks like TriviaQA or Jeopardy which evaluate the exact match of a short-form generation exhibit lower error than the multi-task averages, and exhibit lower noise around the prediction target. For practitioners, we argue using individual tasks may be a better decision in some cases than the multi-task average, if that task better represents the ability than a multi-task average.

Our core results report SNR at the scales of our experimental settings for decision accuracy and prediction error. However, SNR can be calculated at any model size, so we show how the `signal-to-noise` ratio changes for tasks at larger 1B, 7B, 13B and 32B scales in Appendix B.3.

## 5 Improving Predictions by Improving SNR

In this section, we introduce three interventions designed to improve the `signal`, `noise`, or SNR: filtering subtasks by SNR (§5.1), averaging checkpoint scores during a training run (§5.2), measuring language modeling loss over the test set using bits-per-byte (§5.3). In each setup, we show using `signal-to-noise` ratio to intervene on the task improved the resulting error in both prediction settings.

### 5.1 Filtering noisy sub-tasks improves `signal-to-noise ratio`

**Setup.** Many tasks are a macro-average of subtasks. We hypothesize that some subset of subtasks is usually higher quality than the rest of the set, and that the `signal-to-noise` ratio may be an indicator of high quality subtasks. To test this, we first calculate the `signal-to-noise` ratio of each subtask, then rank the subtasks by `signal-to-noise` ratio and greedily add the highest SNR subtasks. As a baseline, we randomly shuffle the subtasks, and report the average of 10 calculations of each metric, with the shading indicating ±1 standard deviation.

**Results.** We show results in Figure 4. For MMLU, using only 16 subtasks had a higher `signal-to-noise` ratio than using the full test set. For AutoBencher, we observe the same but with only 6 tasks. The lower `signal-to-noise` ratio also led to a higher decision accuracy: +2.6% for MMLU and +5% for AutoBencher by using the high SNR subset compared to the full benchmark. We hypothesize that the quality of a task subset may influce that task's `signal-to-noise` ratio. To test this, we use the data collected from MMLU Redux, which identified MMLU subtasks with high labeling error [21]. We find that out of the 20 MMLU subtasks which contain errors in least 5% of instances, half of these subtasks (10 of 20) are also in the lowest 20 tasks sorted by their `signal-to-noise` ratio. This presents

Table 1: Evaluating an intervention designed to average out noise: for a given model on one benchmark, we calculate its score as the average of the scores of its final $k$ checkpoints (evaluated using bits-per-byte task formulation). **Left**: On small models used to make predictions ('Avg. Pred.'), or to the large target models ('Avg. Target'), or both ('Avg. Both'), decision accuracy improves. * indicates the decision accuracy is the same across columns. **Right**: On small models used to fit scaling laws ('Avg. Train'), scaling law error improves. We show results on a subset of benchmarks, and report all benchmarks and the primary metric (accuracy, exact match, pass@1) in Tables 5 and 6.

| Decision Accuracy (60M-5xC to 1B-5xC), % | | | | | Prediction Error (13B-5T), Abs. % | | |
| --- | --- | --- | --- | --- | --- | --- | --- |
| Task ↓ | Final Ckpt | Avg. Pred. | Avg. Target | Avg. Both | Task ↓ | Final Ckpt | Avg. Train |
| **Knowledge QA Tasks** | | | | | **Knowledge QA Tasks** | | |
| ARC Challenge | 94.5 | **94.9** | 94.3 | 94.6 | HellaSwag | 0.31 | **0.16** |
| HellaSwag | 92.4 | 93.1 | 93.1 | **94.0** | CommonsenseQA | 0.59 | **0.46** |
| ARC Easy | 92.1 | **92.2** | 91.9 | 92.0 | Jeopardy | 0.57 | **0.54** |
| MMLU | 91.5 | 91.6 | 91.6 | **91.6** | SocialIQA | **0.50** | 0.59 |
| AutoBencher | 88.5 | 88.9 | 89.1 | **89.6** | PIQA | **0.89** | 1.01 |
| MMLU Pro | **90.0** | 89.4 | 90.0 | 89.3 | MMLU | **1.68** | 1.74 |
| AGI Eval | 86.3 | 86.7 | 86.5 | **87.0** | MMLU Pro | 1.76 | **1.75** |
| MedMCQA* | 86.6 | 86.6 | 86.6 | 86.6 | AGI Eval | **1.89** | 1.98 |
| Jeopardy | 84.4 | 84.4 | 84.8 | **85.0** | BoolQ | 4.13 | **2.48** |
| TriviaQA | 83.5 | 84.3 | 83.8 | **84.6** | TriviaQA | **2.33** | 2.62 |
| OpenBookQA | 81.4 | 81.7 | 81.6 | **82.0** | SQuAD | 2.80 | **2.79** |
| SocialIQA | **79.9** | 79.5 | 79.4 | 79.0 | OpenBookQA | 4.02 | **3.38** |
| PIQA | 72.5 | **72.9** | 71.9 | 72.0 | AutoBencher | 3.86 | **3.69** |
| CommonsenseQA | 65.8 | **66.2** | 65.4 | 65.6 | ARC Easy | 5.13 | **5.13** |
| BoolQ | 63.7 | **64.2** | 63.5 | 64.0 | MedMCQA | **7.72** | 7.98 |
| SQuAD | 60.8 | 60.4 | **62.0** | 61.6 | ARC Challenge | 8.44 | **8.43** |
| Knowledge 19-Task Avg. | 71.3 | 71.5 | **71.7** | **71.7** | Knowledge 19-Task Avg. | 1.43 | **1.20** |
| **Code Tasks** | | | | | **Code Tasks** | | |
| HumanEval* | 95.6 | 95.6 | 95.6 | 95.6 | MBPP | 2.57 | **1.79** |
| MBPP* | 95.3 | 95.3 | 95.3 | 95.3 | HumanEval | **7.71** | 8.85 |
| Code 4-Task Avg.* | 96.7 | 96.7 | 96.7 | 96.7 | Code 4-Task Avg. | 3.15 | **2.75** |
| **Math Tasks** | | | | | **Math Tasks** | | |
| Minerva MATH* | 90.0 | 90.0 | 90.0 | 90.0 | Minerva MATH | 1.08 | **0.98** |
| GSM8K* | 76.6 | 76.6 | 76.6 | 76.6 | GSM8K | 7.46 | **3.85** |
| Math 6-Task Avg.* | 88.3 | 88.3 | 88.3 | 88.3 | Math 6-Task Avg. | 11.33 | **2.30** |
| All 30-Task Avg. | 68.9 | 70.7 | 69.5 | **71.3** | All 30-Task Avg. | 1.03 | **0.86** |

evidence that low SNR may indicate low quality tasks, and we believe this is a good opportunity for future work in evaluation development.

Intuitively, a benchmark developer may increase the statistical power of a comparison between models: by sampling more data by the original process used to construct the benchmark, in order to make a benchmark larger [64], or collect a larger number of tasks in an evaluation suite [58]. Our evidence in Figure 4 suggests that larger benchmarks may not necessarily be better for comparing models. We further explore this phenomenon in App. B.2 by sub-sampling instances of benchmarks, finding some benchmarks can exhibit a higher SNR despite having 10 times fewer instances.

## 5.2 Averaging checkpoint-to-checkpoint `noise` leads to better predictions

**Setup.** Typically, models are only compared using the evaluation of the final checkpoint. In the previous sections, we argued that `noise` is a good indicator of whether we can use a benchmark to predict a large scale phenomenon. In this section, we want to measure the effect of averaging this particular source of step-to-step `noise`, as a way of improving our ability to make a prediction. In the decision accuracy setting, we can average the results of the small model, the large model (in this case, the 1B model), or both. In the prediction error setting, averaging the small models will help in fitting the scaling law, but averaging the target model will just make the result more reliable, so we average the target model in both settings and only change whether we average the models used to fit the scaling law. Finally, we introduce an additional way to average step-to-step noise during a training run, by evaluating whether the ranking of the 1B models during training agrees with the ranking at the end of training. Note, as our measure of `noise` is between intermediate training checkpoints, we are only reducing one of many sources of modeling `noise`.

**Results on Final Checkpoints.** In Table 1, we observe averaging the `noise` improved both measures of error. Averaging `noise` improved decision accuracy by +2.4% for the 30-task average, this procedure improved decision accuracy in all but two tasks. For reducing the scaling law prediction error, averaging the training checkpoints improved prediction error for 20 of 30 tasks.

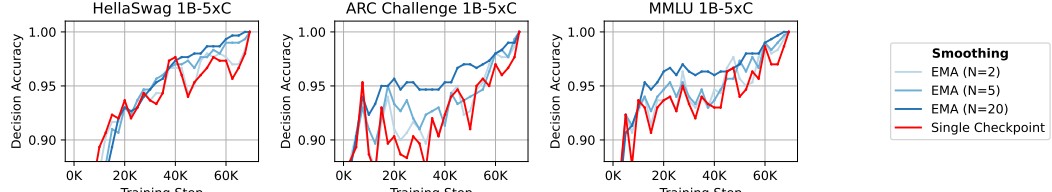

Figure 5: When stopping a training run early, averaging the checkpoint-to-checkpoint noise improves the decision accuracy between an intermediate and the final training step. Shown are decision accuracy from early-stopping for HellaSwag, ARC-C and MMLU by using both a single checkpoint and the exponential moving average (EMA), with all tasks included in Figure 18.

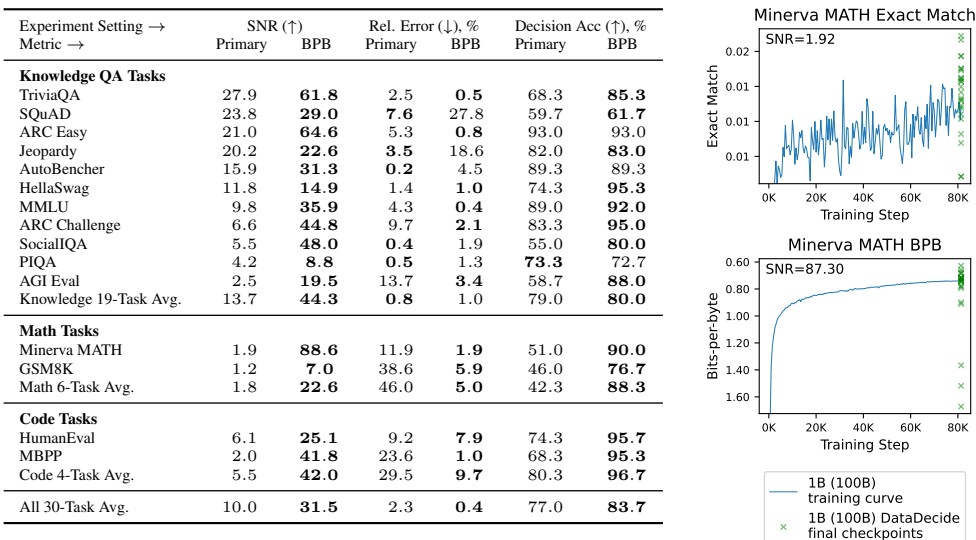

| Experiment Setting → | SNR (↑) | | Rel. Error (↓), % | | Decision Acc (↑), % | |
|---|---|---|---|---|---|---|
| Metric → | Primary | BPB | Primary | BPB | Primary | BPB |
| **Knowledge QA Tasks** | | | | | | |
| TriviaQA | 27.9 | **61.8** | 2.5 | **0.5** | 68.3 | **85.3** |
| SQuAD | 23.8 | **29.0** | 7.6 | 27.8 | 59.7 | **61.7** |
| ARC Easy | 21.0 | **64.6** | 5.3 | **0.8** | 93.0 | 93.0 |
| Jeopardy | 20.2 | **22.6** | 3.5 | 18.6 | 82.0 | **83.0** |
| AutoBencher | 15.9 | **31.3** | 0.2 | 4.5 | 89.3 | 89.3 |
| HellaSwag | 11.8 | **14.9** | 1.4 | **1.0** | 74.3 | **95.3** |
| MMLU | 9.8 | **35.9** | 4.3 | **0.4** | 89.0 | **92.0** |
| ARC Challenge | 6.6 | **44.8** | 9.7 | **2.1** | 83.3 | **95.0** |
| SocialIQA | 5.5 | **48.0** | 0.4 | 1.9 | 55.0 | **80.0** |
| PIQA | 4.2 | **8.8** | 0.5 | 1.3 | **73.3** | 72.7 |
| AGI Eval | 2.5 | **19.5** | 13.7 | **3.4** | 58.7 | **88.0** |
| Knowledge 19-Task Avg. | 13.7 | **44.3** | 0.8 | 1.0 | 79.0 | **80.0** |
| **Math Tasks** | | | | | | |
| Minerva MATH | 1.9 | **88.6** | 11.9 | **1.9** | 51.0 | **90.0** |
| GSM8K | 1.2 | **7.0** | 38.6 | **5.9** | 46.0 | **76.7** |
| Math 6-Task Avg. | 1.8 | **22.6** | 46.0 | **5.0** | 42.3 | **88.3** |
| **Code Tasks** | | | | | | |
| HumanEval | 6.1 | **25.1** | 9.2 | **7.9** | 74.3 | **95.7** |
| MBPP | 2.0 | **41.8** | 23.6 | **1.0** | 68.3 | **95.3** |
| Code 4-Task Avg. | 5.5 | **42.0** | 29.5 | **9.7** | 80.3 | **96.7** |
| All 30-Task Avg. | 10.0 | **31.5** | 2.3 | **0.4** | 77.0 | **83.7** |

Figure 6: Impact of changing benchmark metric to bits-per-byte (BPB) from the primary score (e.g., accuracy, pass@1, etc.). **Left.** Columns are (i) SNR of 1B models trained to 100B tokens; (ii) scaling law prediction error of 1B (and smaller) models used to predict 13B model performance; (iii) decision accuracy for using 150M model to predict 1B model ranking. For almost all tasks at the scales explored here, bits-per-byte shows a higher SNR, and lower scaling law prediction error, and higher decision accuracy than the primary score. Full results across 30 benchmarks and model scales in Table 17. **Right.** Example of primary metric and BPB on a single 1B (100B tokens) training curve (blue curve) and the final checkpoint of 25 models for Minerva MATH (green 'x's). Visually, the BPB training curve is smoother, corresponding to a higher SNR and a lower error in the prediction settings reported in the table, with all tasks in Figure 14.

**Results on Early Stopping.** Another prediction setting is to determine whether the ranking of two partially trained models will exhibit the same order at the end of training. We hypothesize that averaging the step-to-step noise will similarly improve this setting. In Figure 5, we report the decision accuracy for early stopping by using a single checkpoint (red), compared to an exponential moving average of the training curve (blue). We find for almost any training step, applying smoothing led to a higher decision accuracy when comparing models during training. In both settings, reducing the checkpoint-to-checkpoint `noise` allowed a more accurate extrapolation.

## 5.3  Measuring bits-per-byte improves benchmark `signal-to-noise ratio`

**Setup.** Recent work has begun to evaluate by using the test set as a perplexity set, with the intuition that the discontinuous metrics like accuracy or exact match erode the relationship between the language modeling perplexity and the downstream metric [54, 25]. We aim to measure whether the intervention to use a continuous metric improves the `signal-to-noise` ratio and corresponding error. We calculate the bits-per-byte (BPB) using the correct continuations of each test set – the bits-per-byte is the negative log likelihood of the correct answer divided by the number of UTF-8

bytes in the answer string [20, 37]. We compare BPB to the 'primary' task metric (accuracy, exact match, pass@1, etc.) on the `signal-to-noise` ratio, and whether it improves decision-making using decision accuracy from 150M to 1B and reduces the scaling law prediction error at 13B.

**Results.** In Figure 6 we report the `signal-to-noise` ratio, scaling law error and decision accuracy for benchmarks using BPB instead of the primary metric, along with an example training curves for Minerva. Most benchmarks have higher `signal-to-noise` ratio when using the BPB, particularly generative math and code benchmarks like GSM8K (1.2 to 7.0) and MBPP (2.0 to 41.8). To verify this improvement in `signal-to-noise` ratio corresponds to an improvement in our decision-making setups, we observe an improvement in decision accuracy at the small scale for 90.0% of all benchmarks and a lower scaling law prediction error for 73.3% of all benchmarks. We see BPB results in dramatic improvement for tasks that small scale models are not able to accomplish at all, primarily generative tasks. Our results confirm that BPB is a useful metric is both a higher quality development benchmark, particularly for challenging tasks at small scales that do not show above random-chance `signal`.

## 6    Related Work and Discussion

Predicting model behavior at large scales is crucial aspect to language model development, as discussed in the beginning of §2. Noise within evaluation benchmarks is frequently studied as the intrinsic noise of the dataset [2, 7, 40, 6], rather than the noise as a result of differences in the model during training. Closest to our work is Madaan et al. [36], which report a measure of SNR using the benchmark score of a *single* model and noise using 10 seed models, rather than a population of models. We find that the noise of a single model alone, while a useful measure of modeling noise, is not sufficient as a measure of correlation to decision accuracy (§4), and show the step-to-step noise is a cheap alternative to seed noise. Similarly, Kydlíček et al. [29] focus on identifying high quality translations of tasks, but do not focus on decision making. Finally, EvalArena [63] also reports a measure of SNR using the final checkpoints of a small/large model pair (e.g., Llama 3 7B vs. 70B). While statistical measures based on intrinsic noise rather than modeling noise are important indicators of dataset noise, we find that many benchmarks may have low statistical variability but high checkpoint-to-checkpoint noise (such as BoolQ, as observed in Figure 9), which can only be captured with a measure of modeling noise.

Interventions to improve evaluation have been well explored, such as constructing higher quality benchmarks by identifying errors [62, 21], expanding test sets [64], selecting high quality instances from benchmarks [45], or generating entirely new synthetic benchmarks from a model [32]. These works typically justify their decisions using inter-annotator agreement, or a high correlation with the original benchmark. We believe this body of work can benefit from verifying their methods using SNR, rather than noise or reconstruction error alone, to indicate whether the benchmark serves as a useful development tool.

Notably, this scope of our connection between the `signal-to-noise` ratio and predicting large scale phenomena is limited to the two decision accuracy and prediction error settings, and only studies the noise of the model during training. Future work may explore how `signal-to-noise` ratio indicates other small-to-large phenomena [65, 57], and the effects of additional sources of noise on the ability to extrapolate from small-scale experiments, such as from the evaluation configuration [55, 22].

In this work, we identify `signal` and `noise` as a cheap way of estimating whether a benchmark is useful in predicting large-scale phenomena with small scale experiments. We conclude that new benchmark development should use these measures of modeling noise as a guide for building evaluation tools for model developers, and practitioners adopt interventions, such as those introduced in this work, that improve their ability to compare models.

## Acknowledgments and Disclosure of Funding

We would like to thank Pang Wei Koh for feedback on the manuscript; and Dany Haddad, Dirk Groeneveld, Luca Soldaini, Matt Jordan, Oyvind Tafjord, Ronan Le Bras and Saumya Malik for insightful discussions. This material is based upon work supported by the U.S. National Science Foundation under Grant No. 2313998. Any opinions, findings, and conclusions or recommendations expressed in this material are those of the author(s) and do not necessarily reflect the views of the U.S. National Science Foundation. IM is supported by the NSF CSGrad4US Fellowship.

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

# A   Methodology Details

## A.1   Scaling Law Details

Hoffmann et al. [24] models the improvement for larger model training budgets as a power function, proportional to the model parameters $N$ and training tokens $D$, with the exact functional form and prediction setup varying between work [44]. Recent work has begun using the downstream task as the prediction target [17, 19], in this work we follow Bhagia et al. [3] by fitting a scaling law function to the language modeling loss over the correct continuation, then from the task loss to the downstream evaluation. We use the following functional form:

$$L(N, D) = \frac{A}{N^\alpha} + \frac{B}{D^\beta} + E, \quad U(L) = \frac{a}{1 + e^{-k(L-L_0)}} + b \tag{3}$$

We follow the same methodology as Bhagia et al. [3] and use the Huber loss to fit $L(N, D)$ and use a non-linear least squares optimizer to fit $U(L)$. The prediction error is defined as the relative error of the scaling law fit:

$$\text{Prediction Error} = \frac{|\text{Measured Value} - \text{True Value}|}{|\text{True Value}|} \tag{4}$$

## A.2   Decision Accuracy Details

Decision accuracy is one of many rank agreement metrics we could use to show that models trained across pre-training corpora agree at a small scale and a large scale. We present two alternatives here:

**Kendall's $\tau$.** Here, rather than report Kendall's $\tau$, we show it is proportional to decision accuracy. Kendall's $\tau$ is defined as the difference between the concordant pairs $C$ and discordant pairs $D$, divided by the total pairs of models: $\tau = (C - D)/\binom{N}{2}$. We can then rewrite decision accuracy defined only by the number of concordant pairs $C$: decision accuracy $= C/\binom{N}{2}$.

Since we do not allow ties, $C$ and $D$ make up the total number of pairs $\binom{N}{2} = C + D$, we can rewrite decision accuracy as follows:

$$\tau = \frac{C - \left(\binom{N}{2} - C\right)}{\binom{N}{2}} = \frac{2C - \binom{N}{2}}{\binom{N}{2}} = 2 \cdot \frac{C}{\binom{N}{2}} - 1$$
$$= 2 \cdot (\text{decision accuracy}) - 1$$

Therefore, the decision accuracy measure in Magnusson et al. [38] is equivalent to Kendall's $\tau$ modulo a scale and shift.

**Spearman's Rank Correlation.** Kendall's $\tau$ is not sensitive to outliers, and instead we can incorporate the strength of the difference in rank with Spearman's $\rho$: $\rho = 1 - \frac{6\sum d_i^2}{n(n^2-1)}$. This statistic will be more sensitive to large differences in model ranking.

We use decision accuracy in this work for consistency, and to provide a more interpretable metric of rank agreement (for instance, a decision accuracy of 80% indicates that 80% of the pairs of mixes agree between the small scale and large scale). To show that both additional measures of agreement produce similar conclusions, we include correlation with these additional measures of agreement in Table 3.

## A.3   Measures of Modeling `Noise`

**Seed `Noise`.** To measure the `noise` introduced from changing the random seed initialization between training runs, we can compute the standard deviation of the final checkpoint from multiple training runs with different random seeds. To estimate seed `noise`, we train $M$ models using the same configuration, and average the scores over the final $n$ checkpoints of $T$ total training checkpoints to smooth the checkpoint-to-checkpoint `noise`, then compute the standard deviation:

$$\text{Seed Noise}(M) = \sigma(M), \quad M_i = \frac{1}{n} \sum_{j=T-n+1}^{T} U(t_j) \tag{5}$$

**Data Order `noise`.** This is `noise` introduced from changing the order of sampled documents from the training data. We estimate the data order `noise` using the same method as seed `noise`.

**Total Variation.** To measure the checkpoint-to-checkpoint `noise` throughout an entire training run, we measure the total variation of the intermediate training checkpoints on the downstream benchmark. We measure total variation as the average change in metric score across $T$ training checkpoints minus an improvement term:

$$\text{Total Variation} = \frac{1}{T}\sum_{t=1}^{T}|U(t) - U(t-1)| - \frac{1}{T}(U(T) - U(0)) \tag{6}$$

**Checkpoint-to-checkpoint `noise`.** Calculating the above sources of `noise` are either too expensive to estimate at large scales (e.g., training LLMs by varying the random seed) or difficult to run (e.g., evaluating every checkpoint on an LLM training curve). Instead, we propose an estimate measuring only the `noise` of the final $n$ training checkpoints of training:

$$\text{Checkpoint-to-checkpoint Noise} = \sigma\left(\{U(t_j)\}_{j=T-k+1}^{T}\right) \tag{7}$$

### A.3.1 Correlation between Sources of `noise`

To measure the relationship between each source of `noise`, we train 10 1B-5xC models varying the random seed initializations and 10 models varying the data order. In Figure 7, we measure the correlation between the seed `noise`, data order `noise` and total variation against the step-to-step `noise`. Each source of `noise` is highly correlated with the step-to-step `noise` ($R \geq 0.9$ for all measures). While it would be ideal to calculate and reduce all sources of `noise`, seed `noise` and data order `noise` are too expensive to measure (e.g., for large model runs as in Madaan et al. [36]), so only calculating step-to-step `noise` is a reasonable estimate for the modeling `noise`. Thus, we use step-to-step `noise` in as our estimate of the modeling `noise`.

### A.3.2 Selecting the Number of Checkpoints in `Noise`

The noise calculation introduced in Section 3.1 requires selecting some $n$ intermediate checkpoints to estimate the checkpoint-to-checkpoint noise. In this section, we provide guidance on selecting $n$, and discuss its impact on our findings. Increasing the number of intermediate checkpoints $n$ will lead to a less biased estimate of noise. Thus, we can calculate the minimum number of $n$ intermediate checkpoint samples such that the sample noise $s_n$ is a reasonable estimate of the population noise $\sigma$.

We first assume the checkpoint to checkpoint scores are independent and normally distributed (which we observe when computing decision accuracy on intermediate checkpoints in Figure 7). Under this assumption, the ratio between the sample variance and the population variance follows a scaled chi squared distribution: $\frac{(n-1)s_n^2}{\sigma^2} \sim \chi_{n-1}^2$

Therefore we would like to calculate the probability that the sample standard deviation $s_n$ is within one standard deviation of the population standard deviation $\sigma$: $|s_n - \sigma| < \sigma$

We can rewrite this inequality:

$$\left|\frac{s_n}{\sigma} - 1\right| < 1 \Rightarrow 0 < \frac{s_n}{\sigma} < 2$$

And then, can substitute the chi-squared distribution to compute the likelihood w.r.t. $n$:

$$\frac{s_n}{\sigma} \sim \sqrt{\frac{\chi_{n-1}^2}{n-1}} \Rightarrow P\left(\sqrt{\frac{\chi_{n-1}^2}{n-1}} < 2\right) \Rightarrow P\left(\chi_{n-1}^2 < 4(n-1)\right)$$

We can then solve the inequality for the smallest value of $n$ for a particular threshold $\alpha$:

$$P\left(\chi_{n-1}^2 < 4(n-1)\right) > \alpha$$

Solving this inequality numerically with $\alpha = 0.95$ for increasing values of $n$, we find that $n = 9$ provides the smallest sample size such that the probability that the sample standard deviation (the

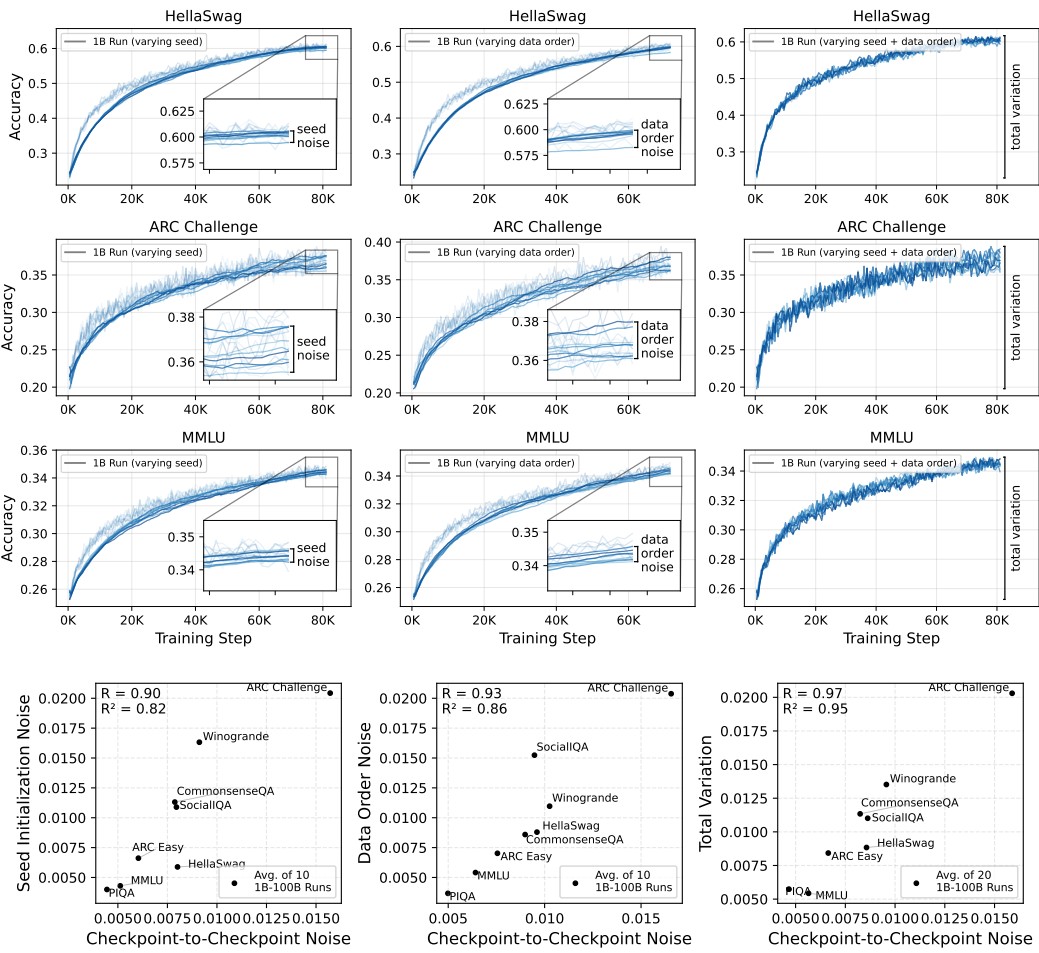

Figure 7: **Top:** 10 different training runs (1B-5×$C$ scale) varying random seed initialization and data order, plotting ARC-C accuracy smoothed across a window of 20 checkpoints. **Bottom:** Total variation or the relative standard deviation (STD normalized by average performance; §3) of scores from different seeds, data after averaging the last 20 training checkpoints vs. the Rel. Std. over the last 20 training checkpoints. Benchmarks with a high checkpoint-to-checkpoint `noise` also exhibit high `noise` due to random seed initialization, data order and `noise` along the full training curve. Noise for all tasks reported in Figure 19.

observed noise) is within one standard deviation of the population standard deviation (the true noise) with 95% confidence. In addition, we can specify a stricter bound by defining the sample standard deviation to be within $k \cdot \sigma$ of the population standard deviation: $|s_n - \sigma| < k \cdot \sigma$

We then verify this empirically using our estimate for noise at the 7B scale (from §5.2). If we assume the 30 intermediate checkpoints provide a reasonable estimate of the population standard deviation, we then compute the sample standard deviation $s_n$ for $n < 30$. We re-compute $s_n$ 1000 times for different subsets to calculate the likelihood that the sampled standard deviation is within $k \cdot \sigma$ of the population standard deviation $\sigma$. In the below table, we report this likelihood with tolerances $k \in \{0.2, 1.0\}$ for subsets $n \in \{5, 10, 20\}$ and bold all results with a likelihood above $0.95$.

In practice, we find that for a large bound ($\pm 1$ std. dev.) can be satisfied for almost all benchmarks with $n = 5$ intermediate checkpoints, but for smaller bounds, (20% of $\pm 1$ std. dev.), using $n = 20$ gives an adequate estimate for 34 of 39 benchmarks we considered in our work.

For our experiment on the 1B-5xC checkpoints, we estimate noise using the average noise of the last 5 checkpoints for all 25 models, so our estimate of noise considers $5 \cdot 25 = 125$ scores.

Table 2: Ablating the $n$ term in `noise`: Likelihood that the sample standard deviation for $n$ intermediate checkpoints is a reasonable estimate for the population standard deviation on OLMo 2 7B, calculated using 30 intermediate checkpoints (Values for $\alpha > 0.95$ in bold). We find that for a low tolerance (within $0.2\sigma$), 20 intermediate checkpoints provides an adequate estimate of noise.

| $k$ threshold in $k \cdot \sigma \rightarrow$ | $k = 0.2$ | | | $k = 1.0$ | | |
| # Ckpts in `Noise` ($n$) $\rightarrow$ | 5 | 10 | 20 | 5 | 10 | 20 |
|---|---|---|---|---|---|---|
| AGI Eval | 0.42 | 0.61 | **0.95** | **1.00** | **1.00** | **1.00** |
| ARC Challenge | 0.44 | 0.70 | **0.98** | **1.00** | **1.00** | **1.00** |
| ARC Easy | 0.38 | 0.65 | **0.97** | **1.00** | **1.00** | **1.00** |
| AutoBencher | 0.47 | 0.71 | **0.97** | **1.00** | **1.00** | **1.00** |
| BBH | 0.42 | 0.60 | 0.95 | **1.00** | **1.00** | **1.00** |
| BoolQ | 0.16 | 0.45 | 0.88 | **1.00** | **1.00** | **1.00** |
| HumanEval | 0.52 | 0.79 | **0.99** | **1.00** | **1.00** | **1.00** |
| HumanEval+ | 0.47 | 0.76 | **0.99** | **1.00** | **1.00** | **1.00** |
| CommonsenseQA | 0.39 | 0.64 | **0.96** | **1.00** | **1.00** | **1.00** |
| DROP | 0.48 | 0.76 | **0.99** | **1.00** | **1.00** | **1.00** |
| GSM8K | 0.49 | 0.77 | **0.99** | **1.00** | **1.00** | **1.00** |
| GSM+ | 0.50 | 0.79 | **0.99** | **1.00** | **1.00** | **1.00** |
| GSM Symbolic | 0.37 | 0.64 | **0.96** | **1.00** | **1.00** | **1.00** |
| GSM Symbolic P1 | 0.47 | 0.69 | **0.98** | **1.00** | **1.00** | **1.00** |
| GSM Symbolic P2 | 0.32 | 0.57 | 0.94 | **1.00** | **1.00** | **1.00** |
| HellaSwag | 0.39 | 0.65 | **0.97** | **1.00** | **1.00** | **1.00** |
| Jeopardy | 0.42 | 0.69 | **0.98** | **1.00** | **1.00** | **1.00** |
| MBPP | 0.43 | 0.63 | **0.96** | **1.00** | **1.00** | **1.00** |
| MBPP+ | 0.41 | 0.63 | **0.96** | **1.00** | **1.00** | **1.00** |
| MedMCQA | 0.50 | 0.79 | **0.99** | **1.00** | **1.00** | **1.00** |
| Minerva MATH | 0.38 | 0.53 | 0.93 | **1.00** | **1.00** | **1.00** |
| Minerva MATH 500 | 0.28 | 0.53 | 0.92 | **1.00** | **1.00** | **1.00** |
| MMLU | 0.00 | 0.00 | 0.54 | 0.83 | **1.00** | **1.00** |
| MMLU Pro | 0.51 | 0.78 | **0.99** | **1.00** | **1.00** | **1.00** |
| All Tasks | 0.00 | 0.00 | 0.08 | 0.83 | **1.00** | **1.00** |
| Code Tasks | 0.49 | 0.78 | **0.99** | **1.00** | **1.00** | **1.00** |
| Knowledge Tasks | 0.00 | 0.00 | 0.15 | 0.83 | **1.00** | **1.00** |
| Math Tasks | 0.55 | 0.83 | **0.99** | **1.00** | **1.00** | **1.00** |
| OLMES Core 9 | 0.31 | 0.49 | 0.92 | **1.00** | **1.00** | **1.00** |
| OLMES Gen | 0.48 | 0.74 | **0.98** | **1.00** | **1.00** | **1.00** |
| OpenBookQA | 0.42 | 0.73 | **0.98** | **1.00** | **1.00** | **1.00** |
| PIQA | 0.43 | 0.69 | **0.98** | **1.00** | **1.00** | **1.00** |
| SocialIQA | 0.30 | 0.44 | 0.88 | 0.99 | **1.00** | **1.00** |
| SQuAD | 0.48 | 0.72 | **0.99** | **1.00** | **1.00** | **1.00** |
| TriviaQA | 0.48 | 0.76 | **0.99** | **1.00** | **1.00** | **1.00** |
| WinoGrande | 0.42 | 0.67 | **0.97** | **1.00** | **1.00** | **1.00** |

## A.4 Measures of `Signal`

**Measurements.** When designing an measure of `signal`, we want to incorporate the uniformity of benchmark scores and the overall range of scores. Given the final checkpoints of training runs under similar compute spend $C_{\text{final}}$, we evaluate multiple approaches to measuring `signal`:

- **Variance** measures average squared distance from the mean: $\text{Var}(C_{\text{final}}) = \frac{1}{n} \sum_{i=1}^{n} \|c_i - \bar{c}\|^2$
- **Mean distance** measures average pairwise distance between points: $\text{Mean Dist}(C_{\text{final}}) = \frac{2}{n(n-1)} \sum_{i<j} \|c_i - c_j\|$
- **Relative standard deviation**, or the coefficient of variation, measures the standard deviation divided by the mean: $\text{Rel. Std.}(C_{\text{final}}) = \frac{\sqrt{\text{Var}(C_{\text{final}})}}{\text{Mean}(C_{\text{final}})}$
- **Star Discrepancy** measures the largest difference between any point and the uniform distribution: $\text{Discrepancy}(C_{\text{final}}) = \sup_{t \in [0,1]} \left| \frac{1}{n} \sum_{i=1}^{n} \mathbf{1}\{c_i \leq t\} - t \right|$.
- **Dispersion** measures the largest difference between any two points, or the largest unfilled space in the range of performance: $\text{Dispersion}(C_{\text{final}}) = \max_{i \neq j} \|c_i - c_j\|$.

Note, we include metrics that are sensitive and non sensitive to outliers, and find our results hold when measuring both types of spread (Table 3). We also include variants of these terms, such using a min-max normalization or scaling by the mean.

**Choosing the a `signal` measurement.** In Table 3, we calculate the correlation between `signal`-to-`noise` ratio and decision accuracy when using each of the `signal` variants. We see that many

Table 3: Correlation of `signal-to-noise` ratio to decision accuracy, using different measures of `signal`. We use the measure which is most predictive of decision accuracy as our measure of `signal`. We include alternative methods for calculating decision accuracy (Pearson correlation and Spearman's rank correlation coefficient), as detailed in Appendix A.2. Fits are illustrated in Figure 10.

| Measure of `Signal` | | SNR vs. Decision Acc $R^2$ | SNR vs. Pearson $R^2$ | SNR vs. Spearman $R^2$ |
|---|---|---|---|---|
| Rel. Dispersion | $\max_{i,j} |c_i - c_j|/\bar{c}$ | 0.5687 | 0.4052 | 0.4902 |
| Rel. Std. Dev. | $\sigma/\mu$ | 0.5657 | 0.3850 | 0.4771 |
| Rel. Mean Pairwise Distance | $\frac{1}{n^2} \sum_{i,j} |c_i - c_j|/\bar{c}$ | 0.5458 | 0.3624 | 0.4561 |
| Interquartile Range | $Q_3 - Q_1$ | 0.4836 | 0.2866 | 0.3980 |
| Distance Standard Deviation | $\frac{1}{n} \sum_i (c_i - \bar{c})$ | 0.4745 | 0.3667 | 0.3950 |
| RMS Deviation | $\sqrt{\frac{1}{n} \sum_i (c_i - \bar{c})^2}$ | 0.4633 | 0.3435 | 0.3812 |
| Mean Pairwise Distance | $\frac{1}{n^2} \sum_{i,j} |c_i - c_j|$ | 0.4589 | 0.3325 | 0.3758 |
| Range | $\max(c) - \min(c)$ | 0.4574 | 0.3604 | 0.3865 |
| Dispersion | $\max_{i,j} |c_i - c_j|$ | 0.4574 | 0.3604 | 0.3865 |
| Quartile Deviation | $(Q_3 - Q_1)/2$ | 0.4528 | 0.2896 | 0.3655 |
| Average Absolute Deviation | $\frac{1}{n} \sum_i |c_i - \bar{c}|$ | 0.4507 | 0.3186 | 0.3672 |
| Median Absolute Deviation | $\text{median}(|c_i - \text{median}(c)|)$ | 0.4168 | 0.2663 | 0.3346 |
| Rel. Mean Squared Pairwise Distance | $\frac{1}{n^2} \sum_{i,j} (c_i - c_j)^2/\bar{c}^2$ | 0.2908 | 0.1627 | 0.2324 |
| Mean Squared Pairwise Distance | $\frac{1}{n^2} \sum_{i,j} (c_i - c_j)^2$ | 0.2480 | 0.1457 | 0.1953 |
| Gini Coefficient | $\frac{1}{2n^2\mu} \sum_{i,j} |c_i - c_j|$ | 0.0944 | 0.0978 | 0.0829 |
| Star Discrepancy (Shift+Scale) | $\sup_{[0,c]} |F_n(t) - F(t)|$ with shifting | 0.0391 | 0.0768 | 0.0454 |
| Star Rel. Discrepancy | $\sup_{[0,c]} |F_n(t) - F(t)|/F(t)$ | 0.0379 | 0.0587 | 0.0420 |
| Dispersion (Shift+Scale) | $\max_{i,j} |c_i - c_j|$ with shifting | 0.0374 | 0.0679 | 0.0382 |
| Halfspace Depth | $\min(F_n(x),\ 1 - F_n(x))$ | 0.0358 | 0.0395 | 0.0373 |
| Discrepancy | $\max_c |F_n(c) - F(c)|$ | 0.0340 | 0.0754 | 0.0401 |
| Projection Depth | $\left(1 + \frac{|x - \text{med}(c)|}{\text{MAD}(c)}\right)^{-1}$ | 0.0331 | 0.0392 | 0.0353 |
| Star Discrepancy | $\sup_{[0,c]} |F_n(t) - F(t)|$ | 0.0319 | 0.0665 | 0.0356 |

straight forward measures have similarly high correlations. We use relative dispersion, the highest correlated among them, as our measure of `signal`.

## A.5 Dataset Details

### A.5.1 Models

We evaluate 465 models which represent stages of the decision-making process during pre-training. Unlike existing collections of model evaluations [18, 33], our set is targeted at development models:

**Scaling Law Models.** 25 ladder models from Bhagia et al. [3]. {190M, 370M, 760M, 1.3B, 3.2B} × {0.5xC, 1xC, 2xC, 5xC, 10xC} trained on OLMoE mix, and 7B-4T / 13B-5T as prediction targets.

**Decision Accuracy Models.** 225 models from Magnusson et al. [38] trained on 25 data recepies for {4M, 20M, 60M, 90M, 150M, 300M, 530M, 750M, 1.3B} trained to 5x Chinchilla optimal.

**Random Seed & Data Order Models.** 20 models 1B-5xC models trained on the OLMoE mix, 10 models trained with different random seed initializations and 10 models trained with different data order seeds.

**Final $n$ Checkpoints.** 120 models representing the 30 final checkpoints before the end of training for OLMo 2 1B, 7B, 13B and 32B [42], with checkpoints spaced by 1000 training checkpoints.

**External Models.** 73 open-weight base models from the DCLM, DeepSeek, Gemma, Llama, Orca, Phi, Pythia, Qwen, SmolLM, StableLM and Yi model families. We estimate the training FLOPs using the reported token count.

We perform all evaluation using up to 2 H100s for a particular model, and use 94K H100 hours total for all evaluation. For training our randomly initialized seed and data order models, we use 23K GPU hours, using a cluster of 2x8 H100s for each training run.

### A.5.2 Benchmarks

We intentionally select benchmarks that are widely adopted in pre-training evaluation. We use the OLMES [22] standard when applicable, and for other benchmarks, we reproduce the evaluation setup

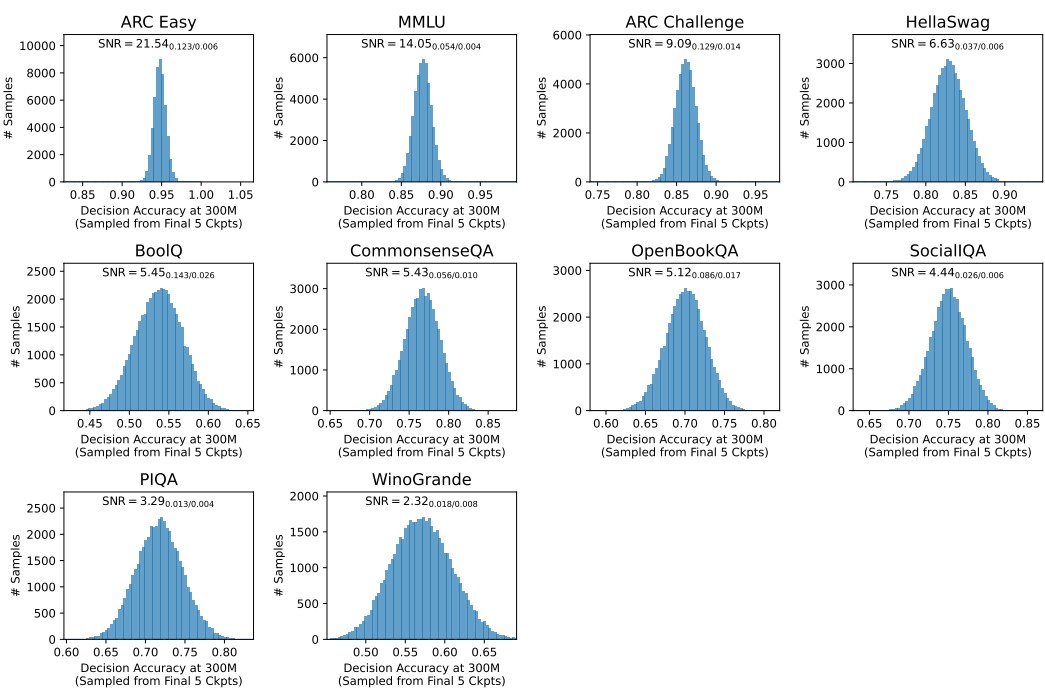

Figure 8: As the benchmark's `signal-to-noise` ratio increases (across histograms), decision accuracy (from 300M to 1B scale) not only increases but becomes more consistent. We test this by resampling decision accuracy for combinations among last 5 checkpoints of the small and large models, respectively, since noise in the results of either size can change rankings. Note how CSQA and MMLU have similar `signal` (Rel. Dispersion $= 0.056$ vs $0.054$) but different `noise` (Rel. Std. $= 0.01$ vs. $0.004$).

from OLMo 2 [42]. Notably, all tasks use few-shot examples and we evaluate MCQA benchmarks in both the rank choice (RC) and multiple choice (MC) setting, since our small ($\leq$1B parameter) models show random-chance performance on MCQA benchmarks.

**Knowledge QA.** MMLU [23], ARC [11], BoolQ [10], CSQA [59], OBQA [39], PiQA [4], SocialIQA [53], HellaSwag [69], WinoGrande [52], DROP [16], CoQA [48], Jeopardy [61], NaturalQs [28], SQuAD [47], TriviaQA [26], MedMCQA [43], MMLU Pro [64], AGI Eval [70], GPQA [49]

**Math.** GSM [12], GSM Plus [46], GSM Symbolic [41], Minerva [30]

**Code.** HumanEval [8], HumanEval+ [34], MBPP [1], MBPP+ [34]

Using strong LLMs have become a tool for augmenting existing benchmarks with more difficult questions or answer choices [64] and re-evaluating benchmark quality [62], and may provide a cheap method for improving `signal`. To test this, we add an additional synthetic benchmark:

**Autobencher.** To test whether fully generated benchmarks can act as an adequate development benchmark, we generate a dataset of 30K MCQA questions using Autobencher [32]. Autobencher iteratively mines for Wikipedia articles and uses a strong LM to generate and prune questions based on saliency, novelty and difficulty constraints.

# B    Full Results

## B.1    Noise measures the reliability of decision accuracy.

As discussed in §3.1, the checkpoint-to-checkpoint `noise` can change the ranking of models, which may effect the decision accuracy we observe by only evaluating the final DataDecide model. To measure the impact of checkpoint-to-checkpoint `noise` on decision accuracy, we can estimate the distribution of possible decision accuracies given the step to step `noise`. To do this, we sample one of the final 5 checkpoints for both the small and large model, and repeatedly sample to estimate the

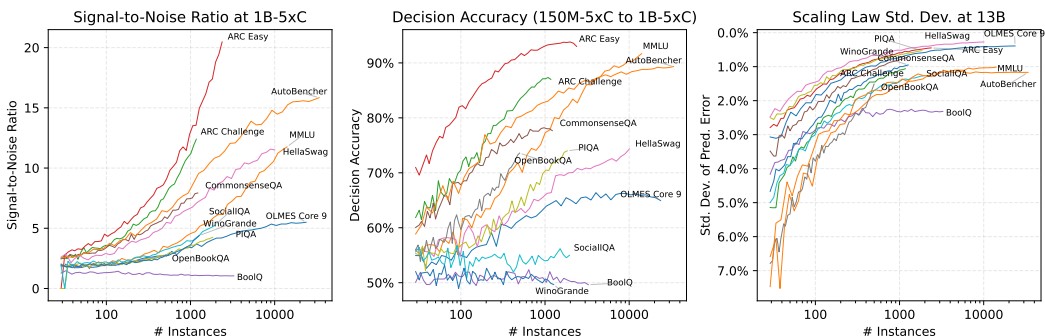

Figure 9: `Signal-to-noise` ratio, decision accuracy, and scaling law prediction error for randomly sampled subsets of instances for 6 development benchmarks. A large sample size alone does not improve `signal-to-noise` ratio. For example, a 1000 question subset of ARC Easy has a higher decision accuracy than MMLU despite having 90% fewer instances.

distribution. A wider distribution would indicate that one should be less confident in the decision accuracy.

We show the distribution of decision accuracies for 10K random samples in Figure 8. For tasks with a higher `signal-to-noise` ratio, the sampled decision accuracy distribution has a higher mean and lower variance. Additionally, we find that tasks with similar `signal`, but different `noise` (e.g., CSQA and MMLU, where CSQA has higher `noise`), the tasks with lower `noise` also have a lower variance of sampled decision accuracy distribution.

### B.2    Increasing benchmark size has diminishing returns

**Setup.** One intuitive way to reduce modeling `noise` is to increase the size of the benchmark, while this is expensive in practice, recent work has given LLMs access to privileged information to generate distractor options or full benchmarks [32, 64]. To test the impact of sample size on modeling `noise`, we use the existing set of benchmarks, select a random sample of instances and recalculate SNR, decision accuracy and scaling law error. To test the limits of synthetic benchmarks, we use our version of AutoBencher, which has 33K instances, or 2x more test instances than the next largest benchmark in our dataset (MMLU).

**Results.** Figure 9 shows how each metric improves as the number of instances increases. Initially, all benchmarks benefit from more samples (up until ∼1K samples) as expected. However, we find dimishing returns for some benchmarks after only 1K instances, in particular the `signal-to-noise` ratio for AutoBencher shows an inflection point at around 2K instances. This is due to the AutoBencher having high `noise`, as shown by the scaling law standard deviation (right figure) – despite having the largest sample size, AutoBencher has the highest checkpoint-to-checkpoint `noise`. In fact, the 300 instance subset of ARC-Easy has lower `noise` than the full 30K instance AutoBench. As using LLMs as part of benchmark construction has become a more popular method of constructing benchmarks, a high quality, small benchmark can actually show a less noisy `signal`.

### B.3    Signal-to-Noise Ratio at Large (>32B) Scales

**Setup.** For models larger than the DataDecide scale (1B-100B), we can rely on the `signal-to-noise` ratio directly to indicate development benchmarks which may not be useful. We estimate the `signal-to-noise` ratio at the compute scales used to train the OLMo 2 models: 1.5B-4T, 7B-4T, 13B-5T and 32B-6T. For `noise`, we use the final 30 intermediate checkpoints, one checkpoint for every 1000 training steps until the end of training. For `signal`, we do not have access to different data recepies trained on the same model, so instead we use a population of open-weight base models trained to similar compute budget as the OLMo 2 models. We use models trained using ±10% of the estimated FLOPs, which results in a population of at least 8 models for each size.

**Results.** Table 4 reports the SNR for each compute budget, sorted by SNR at the 1.5B-4T model scale. SNR can indicate when benchmarks saturated, for example ARC Easy and SocialIQA have high SNR at 1.5B-4T, but low SNR at 32B-6T: 7.89 to 5.10 and 8.73 to 1.95 respectively. For these

Table 4: `Signal-to-noise` ratio for language model development benchmarks for the compute budgets of the OLMo 2 family [42]. For benchmarks measuring a similar ability, we recommend using benchmarks with a higher `signal-to-noise` ratio ratio for a particular model scale. Performance on all models is shown in Figure 12.

| Model Size → | 1.5B-4T | 7B-4T | 13B-5T | 32B-6T |
|---|---|---|---|---|
| Compute → | $2 \cdot 10^{22}$ FLOPs | $1.6 \cdot 10^{23}$ FLOPs | $3.9 \cdot 10^{23}$ FLOPs | $1.2 \cdot 10^{24}$ FLOPs |
| Benchmark ↓ | $\text{SNR}_{\text{Signal/Noise}}$ | $\text{SNR}_{\text{Signal/Noise}}$ | $\text{SNR}_{\text{Signal/Noise}}$ | $\text{SNR}_{\text{Signal/Noise}}$ |
| **Knowledge QA Tasks** | | | | |
| HellaSwag | $39.77_{0.180/0.005}$ | $23.94_{0.061/0.003}$ | $17.81_{0.054/0.003}$ | $8.20_{0.028/0.003}$ |
| TriviaQA | $28.15_{0.411/0.015}$ | $47.03_{0.135/0.003}$ | $60.37_{0.141/0.002}$ | $27.19_{0.064/0.002}$ |
| Jeopardy | $23.66_{0.374/0.016}$ | $14.38_{0.082/0.006}$ | $18.49_{0.084/0.005}$ | $8.00_{0.032/0.004}$ |
| OLMES Gen | $19.34_{0.247/0.013}$ | $32.58_{0.129/0.004}$ | $4.19_{0.092/0.022}$ | $1.06_{0.048/0.046}$ |
| OLMES Core 9 | $19.11_{0.118/0.006}$ | $9.61_{0.039/0.004}$ | $7.13_{0.030/0.004}$ | $8.16_{0.027/0.003}$ |
| AutoBencher | $17.62_{0.264/0.015}$ | $11.42_{0.102/0.009}$ | $8.23_{0.105/0.013}$ | $3.73_{0.050/0.014}$ |
| MMLU Pro | $16.28_{0.246/0.015}$ | $17.44_{0.168/0.010}$ | $9.34_{0.098/0.010}$ | $15.04_{0.136/0.009}$ |
| MMLU | $14.52_{0.139/0.010}$ | $3.39_{0.078/0.023}$ | $7.51_{0.044/0.006}$ | $5.19_{0.061/0.012}$ |
| PIQA | $14.23_{0.058/0.004}$ | $5.31_{0.023/0.004}$ | $5.52_{0.023/0.004}$ | $4.97_{0.015/0.003}$ |
| WinoGrande | $14.12_{0.118/0.008}$ | $7.35_{0.062/0.008}$ | $7.68_{0.070/0.009}$ | $6.60_{0.046/0.007}$ |
| CommonsenseQA | $12.17_{0.120/0.010}$ | $5.66_{0.033/0.006}$ | $2.69_{0.022/0.008}$ | $7.05_{0.039/0.006}$ |
| DROP | $10.79_{0.337/0.031}$ | $20.79_{0.262/0.013}$ | $12.19_{0.226/0.019}$ | $9.01_{0.143/0.016}$ |
| ARC Challenge | $9.41_{0.193/0.021}$ | $5.85_{0.081/0.014}$ | $2.32_{0.033/0.014}$ | $4.74_{0.064/0.014}$ |
| SocialIQA | $8.73_{0.119/0.014}$ | $5.15_{0.049/0.010}$ | $1.69_{0.020/0.012}$ | $1.95_{0.026/0.013}$ |
| MedMCQA | $8.59_{0.106/0.012}$ | $5.79_{0.051/0.009}$ | $7.70_{0.060/0.008}$ | $4.00_{0.041/0.010}$ |
| ARC Easy | $7.89_{0.102/0.013}$ | $5.77_{0.035/0.006}$ | $3.94_{0.018/0.004}$ | $5.10_{0.018/0.004}$ |
| SQuAD | $6.11_{0.090/0.015}$ | $9.76_{0.061/0.006}$ | $10.45_{0.044/0.004}$ | $3.92_{0.027/0.007}$ |
| AGI Eval | $5.31_{0.105/0.020}$ | $4.23_{0.076/0.018}$ | $2.74_{0.050/0.018}$ | $5.40_{0.062/0.012}$ |
| BoolQ | $4.87_{0.116/0.024}$ | $2.99_{0.048/0.016}$ | $1.18_{0.016/0.013}$ | $2.67_{0.016/0.006}$ |
| OpenBookQA | $4.82_{0.145/0.030}$ | $2.13_{0.053/0.025}$ | $2.42_{0.048/0.020}$ | $3.05_{0.063/0.021}$ |
| **Math Tasks** | | | | |
| GSM+ | $8.06_{0.610/0.076}$ | $13.07_{0.500/0.038}$ | $8.55_{0.299/0.035}$ | $8.42_{0.199/0.024}$ |
| GSM Symbolic P1 | $7.18_{0.831/0.116}$ | $4.85_{0.677/0.140}$ | $6.54_{0.450/0.069}$ | $5.31_{0.277/0.052}$ |
| GSM8K | $3.83_{0.587/0.153}$ | $8.21_{0.434/0.053}$ | $6.98_{0.255/0.037}$ | $6.61_{0.160/0.024}$ |
| GSM Symbolic P2 | $3.62_{0.805/0.222}$ | $2.98_{0.769/0.258}$ | $3.39_{0.560/0.165}$ | $4.67_{0.468/0.100}$ |
| GSM Symbolic | $3.05_{0.662/0.217}$ | $8.94_{0.527/0.059}$ | $6.61_{0.283/0.043}$ | $4.29_{0.134/0.031}$ |
| Minerva MATH | $2.28_{0.568/0.250}$ | $9.32_{0.643/0.069}$ | $7.48_{0.567/0.076}$ | $10.19_{0.409/0.040}$ |
| Minerva MATH 500 | $0.91_{0.491/0.539}$ | $4.45_{0.748/0.168}$ | $4.44_{0.647/0.146}$ | $4.30_{0.383/0.089}$ |
| **Code Tasks** | | | | |
| HumanEval+ | $3.70_{0.482/0.130}$ | $7.18_{0.432/0.060}$ | $8.47_{0.377/0.045}$ | $3.34_{0.131/0.039}$ |
| HumanEval | $3.64_{0.452/0.124}$ | $6.25_{0.395/0.063}$ | $5.18_{0.314/0.061}$ | $3.19_{0.117/0.037}$ |
| MBPP+ | $0.88_{0.207/0.235}$ | $3.60_{0.302/0.084}$ | $4.72_{0.265/0.056}$ | $2.94_{0.137/0.047}$ |
| MBPP | $0.88_{0.221/0.251}$ | $5.09_{0.382/0.075}$ | $4.52_{0.255/0.057}$ | $3.57_{0.167/0.047}$ |
| **Multi-task Averages** | | | | |
| Knowledge Tasks | $17.70_{0.146/0.008}$ | $1.61_{0.080/0.049}$ | $9.82_{0.048/0.005}$ | $1.03_{0.058/0.056}$ |
| OLMES + Gen | $17.35_{0.143/0.008}$ | $2.65_{0.074/0.028}$ | $9.52_{0.045/0.005}$ | $0.93_{0.052/0.056}$ |
| All Tasks | $13.92_{0.152/0.011}$ | $3.68_{0.128/0.035}$ | $9.26_{0.055/0.006}$ | $2.94_{0.075/0.026}$ |
| Math Tasks | $5.78_{0.656/0.113}$ | $11.72_{0.580/0.050}$ | $5.06_{0.384/0.076}$ | $7.87_{0.253/0.032}$ |
| Code Tasks | $3.28_{0.333/0.102}$ | $8.20_{0.371/0.045}$ | $8.87_{0.308/0.035}$ | $5.55_{0.126/0.023}$ |

benchmarks, they have less powerful comparisons at larger sizes. SNR also indicates when particular benchmarks become useful. For example, Minerva MATH 500 has the lowest SNR of all tasks at 1.5B-4T (SNR = 0.91) but much higher SNR already at 7B-4T (SNR = 4.45).

Additionally, some individual tasks show better SNR than mutli-task averages. For the OLMES Core 9 average, HellaSwag has higher SNR at all model sizes. For OLMES Gen, TriviaQA has higher SNR at all model sizes. In cases where the SNR of the mutli-task average is low, like the OLMES Average, we recommend comparing models based on individual, high SNR tasks.

## C  Additional Results

We include for our core experiments across all benchmarks we study:

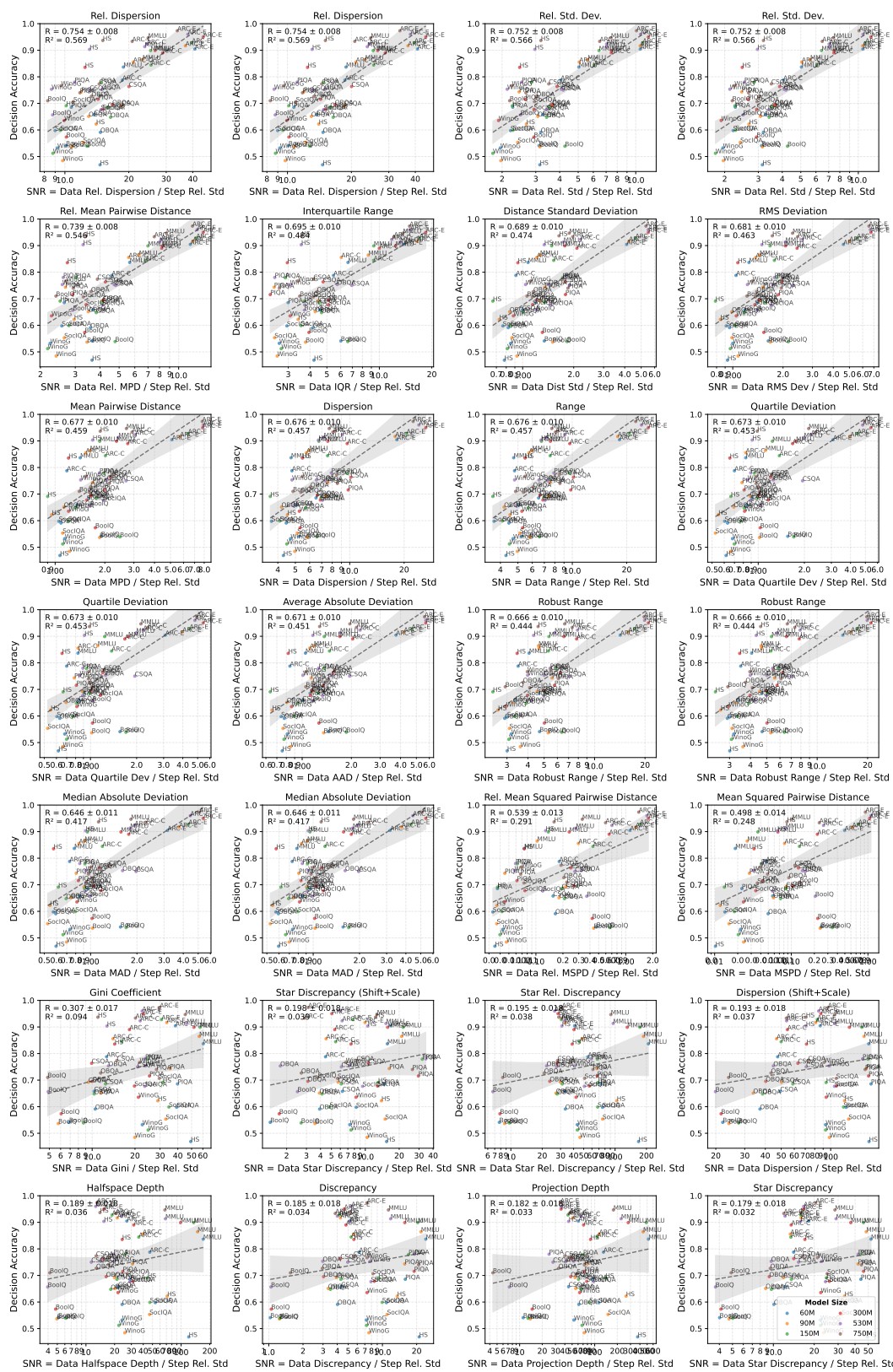

Figure 10: Correlation between decision accuracy and variants of signal-to-noise ratio, using different measures of signal. To pick the measure of signal, we use the metric which is most predictive of decision accuracy.

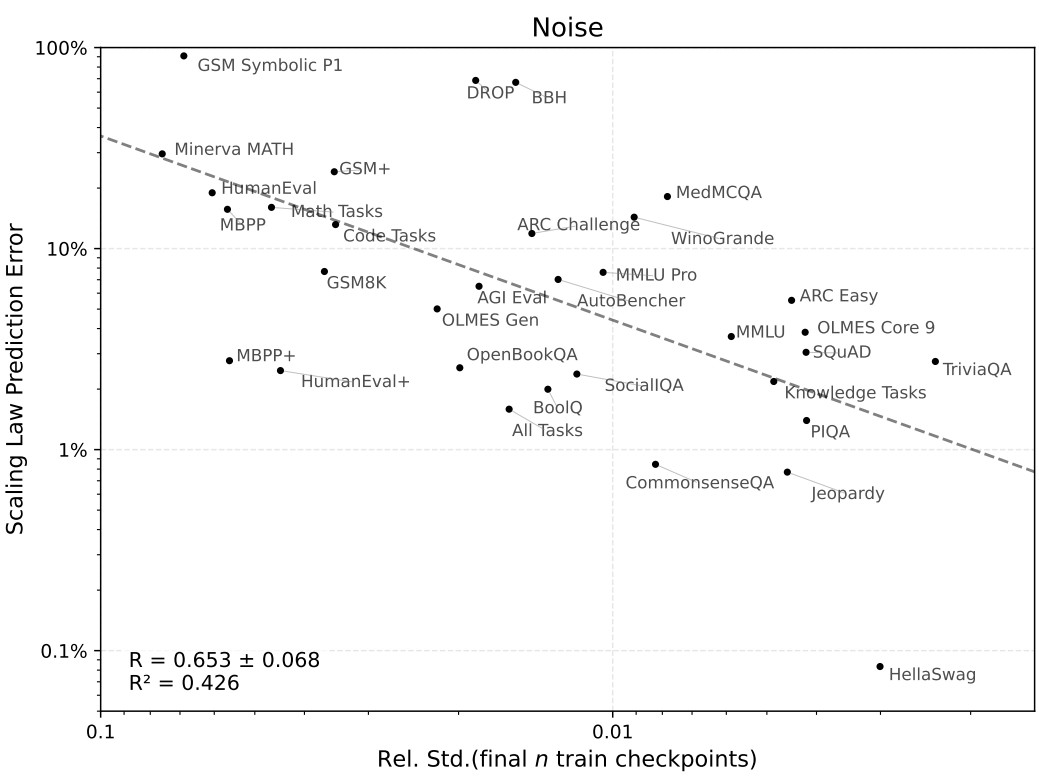

Figure 11: Scaled-up version of the Figure 3 in §4.2 with labels on each task.

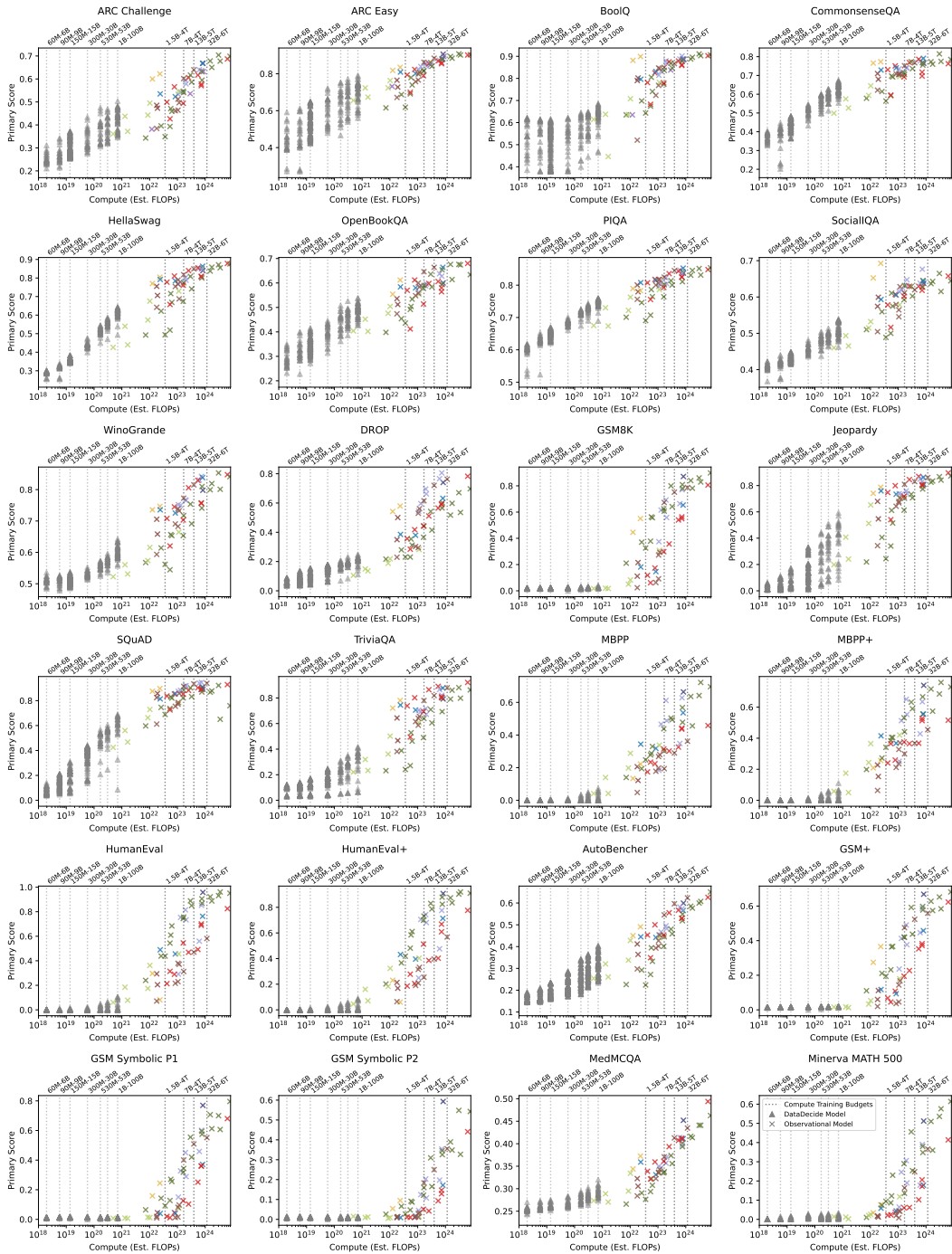

Figure 12: Performance of language models from 60M parameters to 32B parameters, which we use to measure spread at different training budgets in Table 4. For our core experiments, we use the DataDecide models to measures spread, and at large scales, we use external models trained at similar compute budgets.

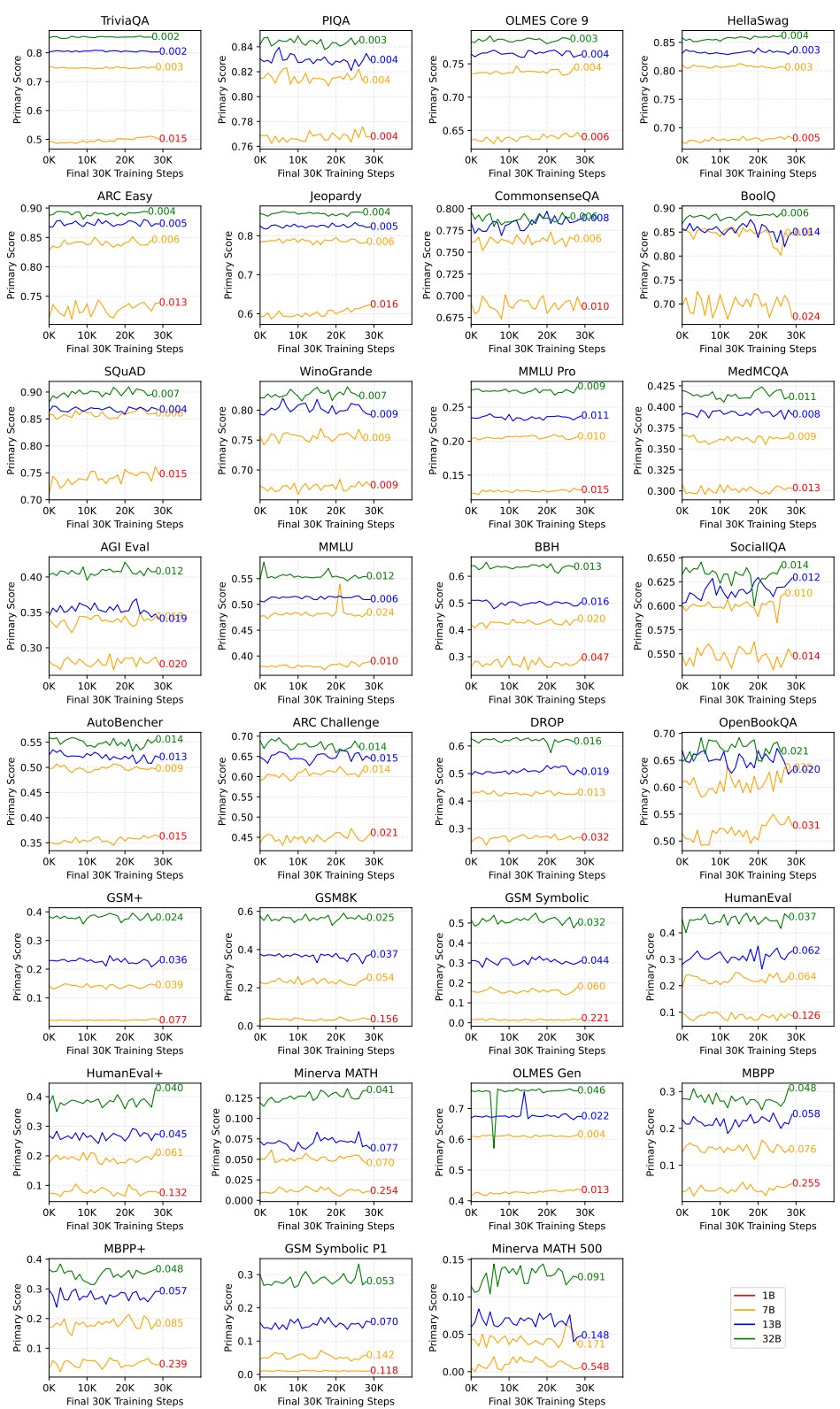

Figure 13: Final 30 checkpoints, each spaced 1000 training steps, for OLMo 2 1B, 7B, 13B and 32B along with the Rel. Std. Dev., which is used to estimate `noise`.

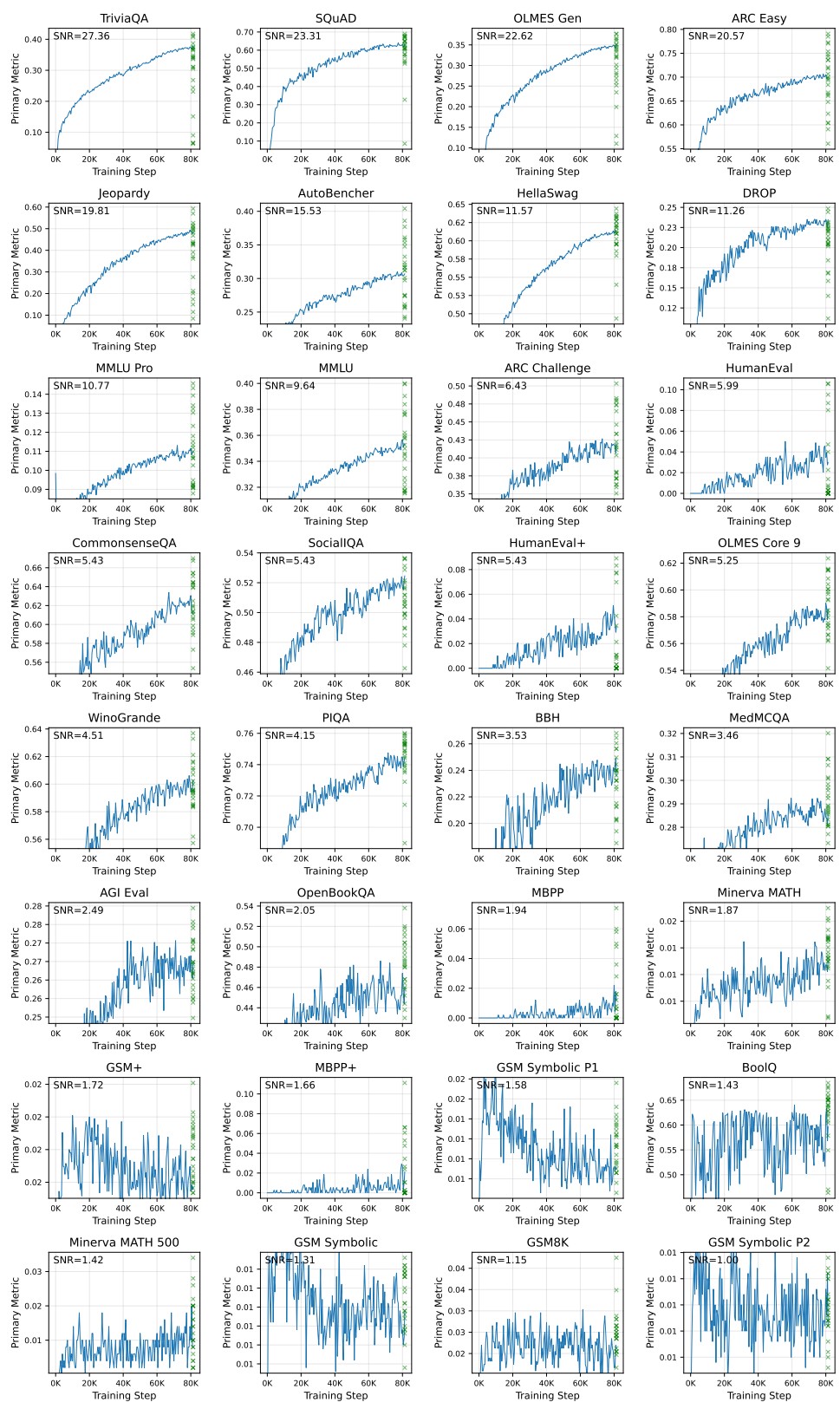

Figure 14: 1B-5xC training curves and final checkpoints for DataDecide models across tasks, sorted by the `signal-to-noise` ratio.

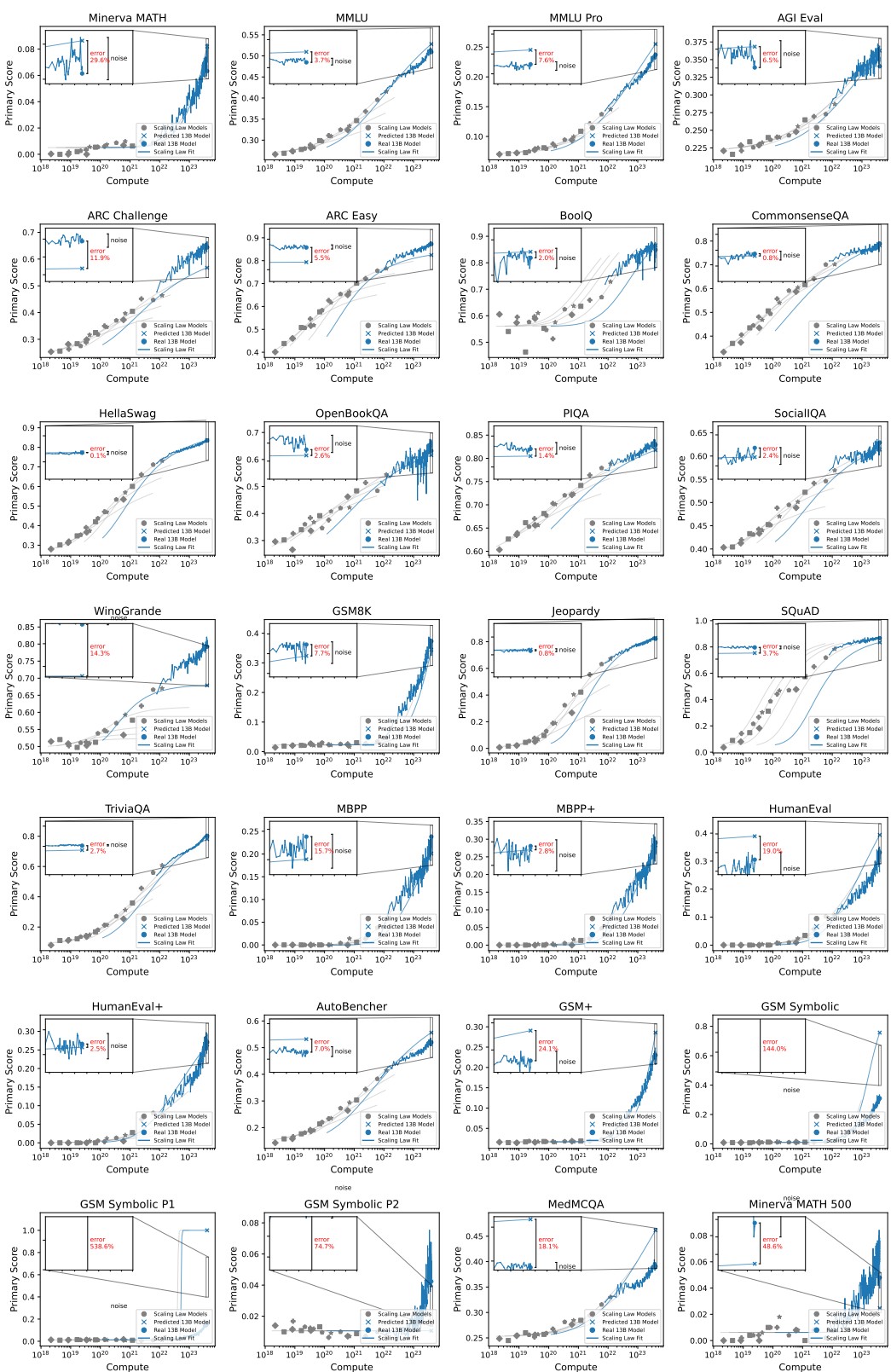

Figure 15: Scaling law fits for all tasks using the OLMo 2 13B-5T prediction target.

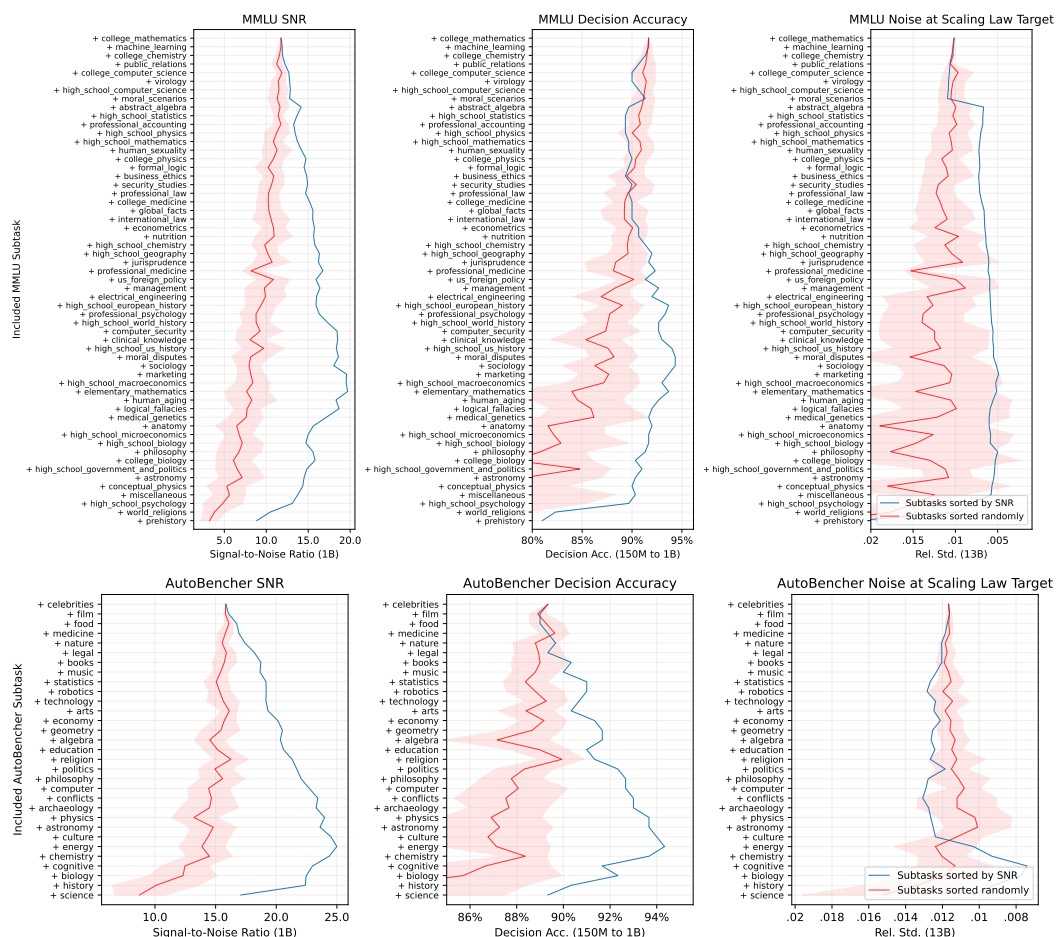

Figure 16: Larger version of Figure 4, showing the names of each subtask, sorted by SNR from bottom (highest SNR) to top (lowest SNR).

Table 5: Scaling law fit error for BPB and primary score for all tasks with averaging the final 5 checkpoints in the ladder train models.

| Task (↓) | Predicting Bits-per-byte | | | | Predicting Primary Score | | | |
| | Abs. Error, % | | Rel. Error, % | | Abs. Error, % | | Rel. Error, % | |
| | Final Only | Avg. Train | Final Only | Avg. Train | Final Only | Avg. Train | Final Only | Avg. Train |
|---|---|---|---|---|---|---|---|---|
| **Knowledge QA Tasks** | | | | | | | | |
| HellaSwag | 0.76 | 0.80 | 1.16 | 1.22 | 0.31 | 0.16 | 0.37 | 0.20 |
| CommonsenseQA | 6.24 | 5.32 | 8.75 | 7.46 | 0.59 | 0.46 | 0.75 | 0.58 |
| Jeopardy | 5.08 | 5.14 | 18.51 | 18.73 | 0.57 | 0.54 | 0.69 | 0.66 |
| SocialIQA | 0.66 | 0.41 | 0.74 | 0.46 | 0.50 | 0.59 | 0.80 | 0.95 |
| PIQA | 1.23 | 1.39 | 1.40 | 1.59 | 0.89 | 1.01 | 1.08 | 1.22 |
| MMLU | 0.56 | 0.49 | 0.75 | 0.66 | 1.68 | 1.74 | 3.28 | 3.39 |
| MMLU Pro | 0.78 | 0.71 | 0.73 | 0.67 | 1.76 | 1.75 | 7.51 | 7.45 |
| AGI Eval | 2.79 | 2.66 | 3.33 | 3.18 | 1.89 | 1.98 | 5.43 | 5.70 |
| OLMES Gen | 4.66 | 2.32 | 3.92 | 1.95 | 4.19 | 2.16 | 6.22 | 3.20 |
| BoolQ | 1.49 | 1.76 | 8.54 | 10.11 | 4.13 | 2.48 | 4.91 | 2.96 |
| OLMES Core 9 | 0.47 | 0.25 | 0.62 | 0.33 | 2.47 | 2.62 | 3.23 | 3.42 |
| TriviaQA | 1.56 | 2.05 | 2.27 | 2.98 | 2.33 | 2.62 | 2.89 | 3.25 |
| SQuAD | 4.96 | 4.96 | 32.35 | 32.37 | 2.80 | 2.79 | 3.23 | 3.21 |
| OpenBookQA | 3.18 | 3.92 | 2.80 | 3.46 | 4.02 | 3.38 | 6.22 | 5.22 |
| AutoBencher | 2.92 | 2.78 | 4.70 | 4.49 | 3.86 | 3.69 | 7.47 | 7.14 |
| ARC Easy | 1.36 | 1.37 | 2.89 | 2.90 | 5.13 | 5.13 | 5.87 | 5.87 |
| MedMCQA | 5.07 | 5.38 | 5.35 | 5.67 | 7.72 | 7.98 | 19.72 | 20.41 |
| ARC Challenge | 2.08 | 2.07 | 3.15 | 3.14 | 8.44 | 8.43 | 13.02 | 13.01 |
| WinoGrande | 1.01 | 1.38 | 0.83 | 1.12 | 10.01 | 10.82 | 12.47 | 13.49 |
| BBH | 61.84 | 65.01 | 12.81 | 13.47 | 33.09 | 33.08 | 66.61 | 66.59 |
| DROP | 47.51 | 48.19 | 10.75 | 10.91 | 35.17 | 35.20 | 68.77 | 68.82 |
| Knowledge 19-Task Avg. | 1.18 | 0.87 | 1.32 | 0.98 | 1.43 | 1.20 | 2.22 | 1.85 |
| **Math Tasks** | | | | | | | | |
| Minerva MATH | 0.73 | 0.66 | 1.50 | 1.36 | 1.08 | 0.98 | 15.28 | 13.93 |
| Minerva MATH 500 | 0.34 | 0.14 | 0.71 | 0.29 | 17.35 | 1.78 | 306.18 | 31.36 |
| GSM Symbolic P2 | 2.57 | 2.83 | 5.23 | 5.75 | 7.46 | 3.50 | 164.53 | 77.13 |
| GSM8K | 2.43 | 2.48 | 5.90 | 6.01 | 7.46 | 3.85 | 20.55 | 10.61 |
| GSM+ | 2.02 | 1.95 | 4.54 | 4.40 | 29.14 | 28.54 | 130.01 | 127.36 |
| GSM Symbolic | 1.87 | 1.71 | 4.64 | 4.25 | 39.88 | 38.88 | 132.62 | 129.30 |
| GSM Symbolic P1 | 2.31 | 2.35 | 5.04 | 5.11 | 27.15 | 83.62 | 178.46 | 549.63 |
| Math 6-Task Avg. | 2.05 | 2.01 | 4.52 | 4.42 | 11.33 | 2.30 | 65.52 | 13.28 |
| **Code Tasks** | | | | | | | | |
| HumanEval+ | 1.92 | 2.21 | 3.57 | 4.10 | 1.05 | 0.04 | 3.91 | 0.16 |
| MBPP | 0.30 | 0.32 | 0.46 | 0.48 | 2.57 | 1.79 | 11.63 | 8.10 |
| MBPP+ | 6.49 | 6.62 | 12.56 | 12.81 | 9.08 | 8.79 | 33.14 | 32.11 |
| HumanEval | 1.59 | 2.01 | 3.85 | 4.87 | 7.71 | 8.85 | 24.00 | 27.55 |
| Code 4-Task Avg. | 3.23 | 3.33 | 6.07 | 6.25 | 3.15 | 2.75 | 11.61 | 10.15 |
| All 30-Task Avg. | 0.47 | 0.15 | 0.62 | 0.20 | 1.03 | 0.86 | 2.10 | 1.76 |

Table 6: Decision accuracy averaging the final 5 checkpoints for bits-per-byte and the primary metric (accuracy, exact match, pass@1).

| Task (↓) | Bits-per-byte, % | | | | Primary Metric, % | | | |
|---|---|---|---|---|---|---|---|---|
| | Final Ckpt | Avg. Pred | Avg. Target | Avg. Both | Final Ckpt | Avg. Pred | Avg. Target | Avg. Both |
| **Knowledge QA Tasks** | | | | | | | | |
| ARC Challenge | 94.56 | **94.88** | 94.38 | 94.67 | 82.91 | 82.27 | **82.91** | 82.00 |
| HellaSwag | 92.42 | 93.19 | 93.21 | **94.00** | 71.05 | 71.26 | **72.37** | 72.33 |
| ARC Easy | **92.23** | 92.15 | 91.96 | 92.00 | 93.96 | 93.99 | **94.05** | 94.00 |
| MMLU | 91.53 | 91.64 | 91.63 | **91.67** | 89.08 | 88.84 | **89.60** | 89.00 |
| AutoBencher | 88.55 | 88.95 | 89.19 | **89.67** | 88.80 | **89.05** | 88.81 | 89.00 |
| MMLU Pro | 90.00 | 89.40 | **90.04** | 89.33 | 83.34 | 83.77 | 84.20 | **84.67** |
| AGI Eval | 86.38 | 86.75 | 86.54 | **87.00** | 57.38 | **58.60** | 56.45 | 57.67 |
| MedMCQA | 86.67 | 86.67 | 86.67 | 86.67 | **61.33** | **61.33** | **61.33** | 60.33 |
| Jeopardy | 84.42 | 84.46 | 84.88 | **85.00** | 83.01 | 82.60 | **83.74** | 83.33 |
| TriviaQA | 83.55 | 84.29 | 83.86 | **84.67** | 69.10 | **69.54** | 69.09 | 69.33 |
| OpenBookQA | 81.53 | 81.75 | 81.68 | **82.00** | 66.82 | 66.98 | 68.05 | **68.33** |
| OLMES Core 9 | 79.05 | 80.10 | 79.32 | **80.33** | **74.67** | 73.92 | 74.24 | 73.67 |
| SocialIQA | **79.92** | 79.57 | 79.45 | 79.00 | 55.58 | 55.58 | 56.09 | **56.67** |
| WinoGrande | 73.20 | **74.29** | 72.83 | 74.00 | 50.52 | 50.27 | 49.81 | 49.00 |
| PIQA | 72.60 | **72.91** | 71.93 | 72.00 | 72.78 | 72.66 | **73.09** | 72.33 |
| CommonsenseQA | 65.86 | **66.25** | 65.42 | 65.67 | 68.74 | 69.05 | 70.61 | **71.00** |
| BoolQ | 63.72 | **64.19** | 63.51 | 64.00 | 50.38 | 48.90 | **50.66** | 49.33 |
| SQuAD | 60.93 | 60.59 | **62.02** | 61.67 | 58.69 | 58.35 | **59.72** | 59.33 |
| OLMES Gen | **61.16** | 55.44 | 55.11 | 58.86 | **62.06** | 54.87 | 53.42 | 50.12 |
| DROP | 56.67 | 56.48 | **57.46** | 57.33 | 57.77 | 59.06 | 57.80 | **59.33** |
| BBH | 57.48 | 57.25 | **57.66** | 57.33 | 59.15 | 59.88 | 60.85 | **61.33** |
| Knowledge 19-Task Avg. | 71.39 | 71.49 | 71.62 | **71.67** | 70.70 | 75.82 | 72.65 | **78.00** |
| **Math Tasks** | | | | | | | | |
| Minerva MATH 500 | 90.33 | 90.33 | 90.33 | 90.33 | 51.00 | 51.00 | 51.00 | 51.00 |
| Minerva MATH | 90.00 | 90.00 | 90.00 | 90.00 | 51.00 | 51.00 | 51.00 | 51.00 |
| GSM Symbolic P1 | 81.33 | 81.33 | 81.33 | 81.33 | 41.67 | 41.67 | 41.67 | 41.67 |
| GSM Symbolic P2 | 79.67 | 79.67 | 79.67 | 79.67 | 40.33 | 40.33 | 40.33 | 40.33 |
| GSM+ | 79.00 | 79.00 | 79.00 | 79.00 | 59.67 | 59.67 | 59.67 | 59.67 |
| GSM Symbolic | 78.33 | 78.33 | 78.33 | 78.33 | 51.67 | 51.67 | 51.67 | 51.67 |
| GSM8K | 76.67 | 76.67 | 76.67 | 76.67 | 46.33 | 46.33 | 46.33 | 46.33 |
| Math 6-Task Avg. | 88.33 | 88.33 | 88.33 | 88.33 | 42.67 | 42.67 | 42.67 | 42.67 |
| **Code Tasks** | | | | | | | | |
| HumanEval+ | 96.33 | 96.33 | 96.33 | 96.33 | 71.33 | 71.33 | 71.33 | 71.33 |
| HumanEval | 95.67 | 95.67 | 95.67 | 95.67 | 80.00 | 80.00 | 80.00 | 80.00 |
| MBPP | 95.33 | 95.33 | 95.33 | 95.33 | 76.00 | 76.00 | 76.00 | 76.00 |
| MBPP+ | 93.00 | 93.00 | 93.00 | 93.00 | 70.67 | 70.67 | 70.67 | 70.67 |
| Code 4-Task Avg. | 96.67 | 96.67 | 96.67 | 96.67 | 85.67 | 85.67 | 85.67 | 85.67 |
| All 30-Task Avg. | 68.57 | 70.63 | 69.78 | **71.33** | 62.15 | 68.88 | 67.29 | **77.33** |

Figure 17: Bits-per-byte vs. primary metric on the full suite of tasks shown in Figure 6.

| Experiment Setting → | SNR (↑) | | Rel. Error (↓), % | | Decision Acc (↑), % | |
| Metric → | Primary | BPB | Primary | BPB | Primary | BPB |
|---|---|---|---|---|---|---|
| **Knowledge QA Tasks** | | | | | | |
| TriviaQA | 27.9 | **61.8** | 2.5 | **0.5** | 68.3 | **85.3** |
| SQuAD | 23.8 | **29.0** | **7.6** | 27.8 | 59.7 | **61.7** |
| OLMES Gen | **23.1** | 20.6 | **0.9** | 2.6 | 63.3 | **67.3** |
| ARC Easy | 21.0 | **64.6** | 5.3 | **0.8** | 93.0 | 93.0 |
| Jeopardy | 20.2 | **22.6** | **3.5** | 18.6 | 82.0 | **83.0** |
| AutoBencher | 15.9 | **31.3** | **0.2** | 4.5 | 89.3 | 89.3 |
| HellaSwag | 11.8 | **14.9** | 1.4 | **1.0** | 74.3 | **95.3** |
| DROP | **11.5** | 9.9 | 59.0 | **11.3** | 57.3 | **58.7** |
| OLMES + Gen | 11.2 | **40.0** | 2.1 | **0.4** | 89.0 | 89.0 |
| MMLU Pro | 11.0 | **27.6** | 2.7 | **1.3** | 83.0 | **89.0** |
| MMLU | 9.8 | **35.9** | 4.3 | **0.4** | 89.0 | **92.0** |
| ARC Challenge | 6.6 | **44.8** | 9.7 | **2.1** | 83.3 | **95.0** |
| CommonsenseQA | 5.5 | **41.9** | **3.6** | 5.9 | **68.7** | 65.7 |
| SocialIQA | 5.5 | **48.0** | 0.4 | **1.9** | 55.0 | **80.0** |
| OLMES Core 9 | 5.4 | **73.2** | 3.7 | **0.2** | 73.3 | **79.3** |
| WinoGrande | **4.6** | 3.6 | 10.3 | **0.9** | 49.7 | **75.0** |
| PIQA | 4.2 | **8.8** | **0.5** | 1.3 | **73.3** | 72.7 |
| BBH | **3.6** | 2.5 | 67.1 | **12.9** | **64.7** | 55.0 |
| MedMCQA | 3.5 | **29.5** | 8.8 | **4.6** | 60.3 | **86.7** |
| AGI Eval | 2.5 | **19.5** | 13.7 | **3.4** | 58.7 | **88.0** |
| OpenBookQA | 2.1 | **24.2** | 7.7 | **3.3** | 65.7 | **82.7** |
| BoolQ | 1.5 | **64.8** | **5.1** | 6.6 | 47.7 | **62.3** |
| Knowledge 19-Task Avg. | 13.7 | **44.3** | **0.8** | 1.0 | 79.0 | **80.0** |
| **Math Tasks** | | | | | | |
| Minerva MATH | 1.9 | **88.6** | 11.9 | **1.9** | 51.0 | **90.0** |
| GSM+ | 1.8 | **7.3** | 20.0 | **4.8** | 59.7 | **79.0** |
| GSM Symb. | 1.3 | **6.5** | 83.0 | **5.1** | 51.0 | **78.3** |
| GSM8K | 1.2 | **7.0** | 38.6 | **5.9** | 46.0 | **76.7** |
| Math 6-Task Avg. | 1.8 | **22.6** | 46.0 | **5.0** | 42.3 | **88.3** |
| **Code Tasks** | | | | | | |
| HumanEval | 6.1 | **25.1** | 9.2 | **7.9** | 74.3 | **95.7** |
| HumanEval+ | 5.5 | **27.4** | 29.7 | **7.1** | 66.0 | **96.3** |
| MBPP | 2.0 | **41.8** | 23.6 | **1.0** | 68.3 | **95.3** |
| MBPP+ | 1.7 | **30.8** | 39.5 | **8.9** | 62.7 | **93.0** |
| GSM Symb. P1 | 1.6 | **6.6** | 538.6 | **5.2** | 41.3 | **81.3** |
| Minerva MATH 500 | 1.4 | **90.5** | 52.5 | **0.9** | 50.7 | **90.3** |
| GSM Symb. P2 | 1.0 | **7.0** | 74.8 | **5.1** | 40.3 | **79.7** |
| Code 4-Task Avg. | 5.5 | **42.0** | 29.5 | **9.7** | 80.3 | **96.7** |
| All 30-Task Avg. | 10.0 | **31.5** | 2.3 | **0.4** | 77.0 | **83.7** |

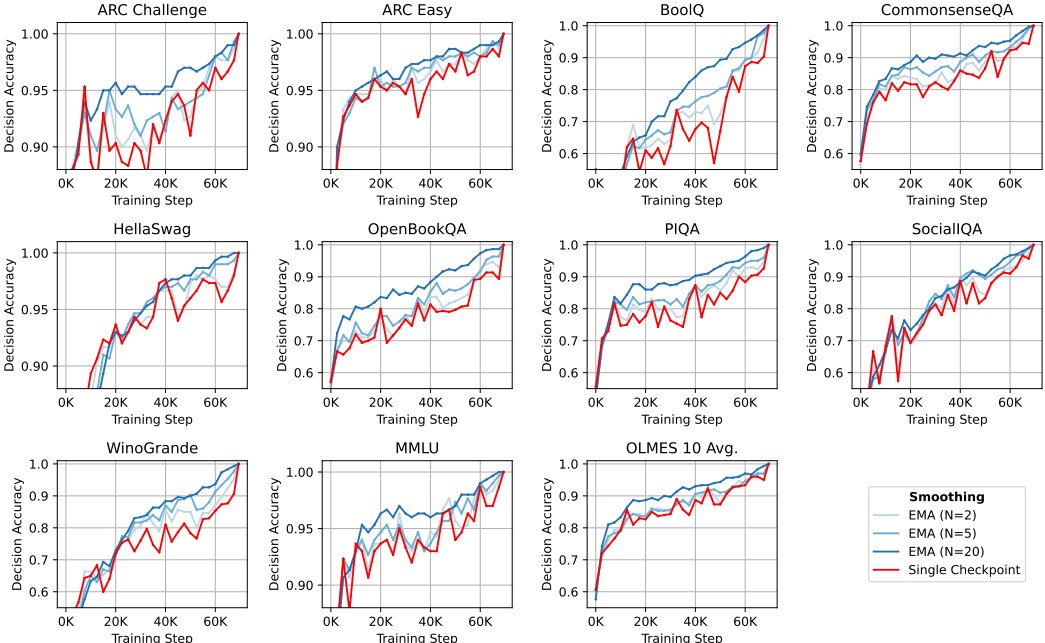

Figure 18: When stopping a training run early, averaging the checkpoint-to-checkpoint noise improves the decision accuracy between an intermediate and the final training step. Shown are decision accuracy from early-stopping for the core OLMES tasks by using both a single checkpoint and the exponential moving average (EMA)

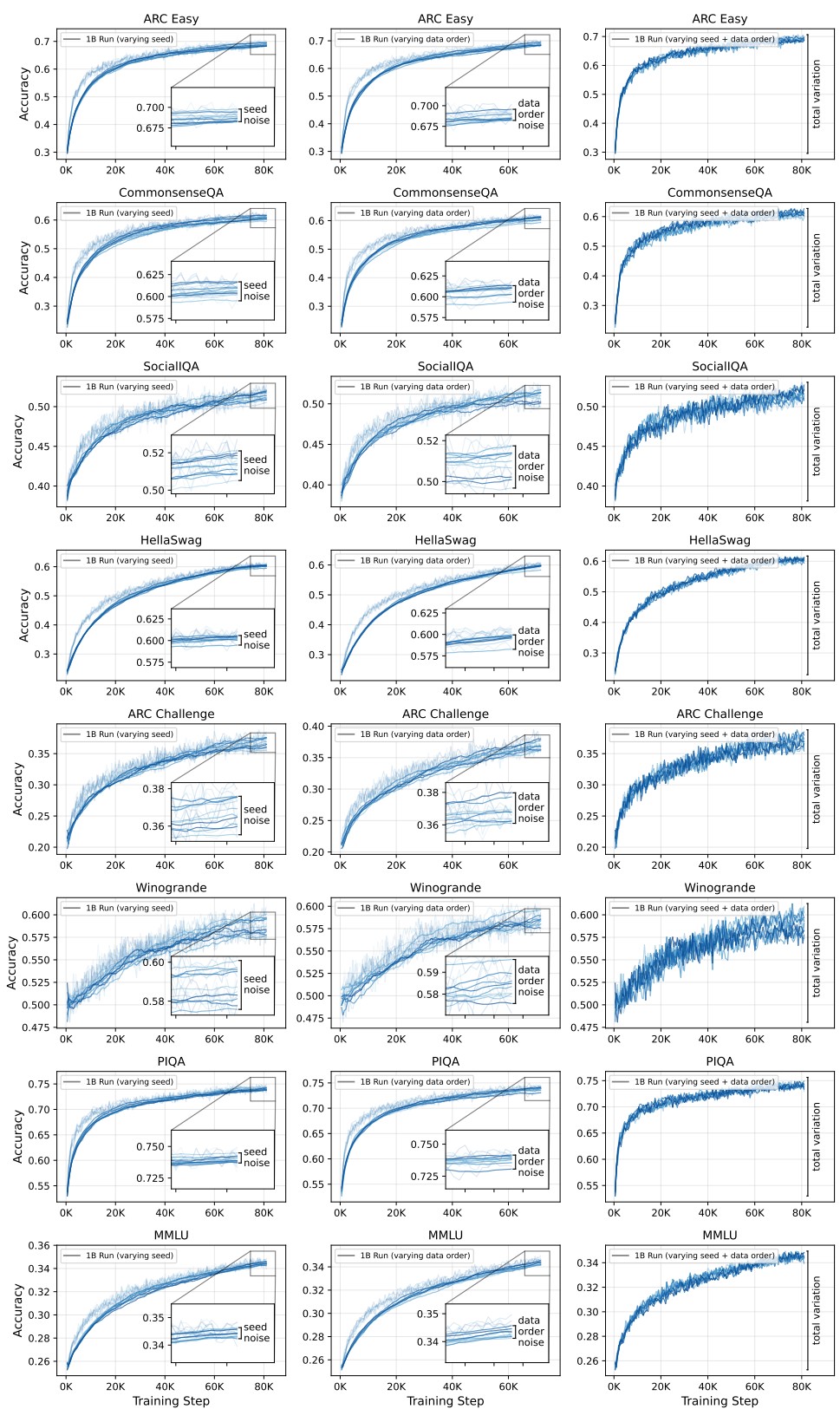

Figure 19: Visualization for the seed `noise`, data order `noise` and total variation for all OLMES tasks.

