# OpenReview forum: "Signal and Noise: A Framework for Reducing Uncertainty in Language Model Evaluation"
_NeurIPS.cc/2025/Conference — NeurIPS 2025 spotlight_

### Official Review · Reviewer_Nqe6 · 2025-06-23

**Clarity:** 2
**Significance:** 3
**Originality:** 3
**Rating:** 5
**Confidence:** 3

**Summary:**

The paper identifies two metrics, signal and noise, to evaluate the reliability and usefulness of benchmarks in decision making during LLM deployment, e.g., model architecture, training approach and training dataset. Signal measures the ability of the benchmark to distinguish different models, while noise measures the sensitivity to randomness during training. The paper discovers that signal to noise ratio (SNR) is highly correlated with decision accuracy. Finally, the paper discusses ways to improve SNR.

**Questions:**

1. How to choose $n$ in Equation (2)? A practical guidance on choosing $n$ greatly improves the significance of the work.
2. SNR is dependent on the models. If a benchmark has high SNR on a set of models $\mathcal M$, will it be reliable and useful for decision making on a new model $m$ that is not in $\mathcal M$?

**Ethical Concerns:**

["NO or VERY MINOR ethics concerns only"]

**Final Justification:**

The authors perfectly resolved the issue of the selection of $n$, and promised to resolve the notation problems in the final version. Therefore, I agree to raise the score to 5.

**Limitations:**

Yes

**Paper Formatting Concerns:**

No issues

**Quality:**

4

**Strengths And Weaknesses:**

**Strength in Quality**: The paper conducts abundant experiments to support the claim. In Section 4, the experiment is conducted on 25 models and three tasks, so the results are convincing.

**Strength in Significance**: The paper contributes to benchmark design by introducing a practical metric. SNR is easy to compute as it only involves small models. Moreover, it is highly correlated to decision accuracy.

**Weakness in Clarity**: The notations are not clearly defined. For instance, in Equation (1), the range of $B(\cdot)$ is not given (i.e., does it output a scalar or a vector or a model?), and $m_a$, $m_b$ are not defined (Though $m$ is defined as a model, what does the subscript mean here?) Also, in line 146, and 160, both use the subscript $i$, but the former refers to training step, while the latter refers to model.

**Weakness in Significance**: SNR computation (Equation (2)) relies on a hyperparameter $n$, which is set differently in experiments (5 in Section 4.1, 30 in Section 4.2). In practice, it is hard to determine the best choice of $n$.

---

> ### Author Rebuttal · Authors · 2025-07-31
>
> Thank you for your review and highlighting the “abundant experiments” and that “the results are convincing.”
>
> Your feedback brought up three points: practical guidance on selecting $n$ in the noise term of SNR, clarifying aspects of our notation, and the impact of the particular model set $\mathcal{M}$ on SNR. We address these points, in that order, below:
>
> **Concerns:**
>
> 1. **Selecting $n$ in the noise term**
>
> Thank you for raising this question. Below, we provide guidance on the selection of $n$ and will include this discussion in the final draft of our work.
>
> In the noise calculation (Eq. 2), increasing the number of intermediate checkpoints $n$ will lead to a less biased estimate of noise. Thus, we can calculate the minimum number of $n$ intermediate checkpoint samples such that the sample noise $s_n$ is a reasonable estimate of the population noise $\sigma$:
>
> We first assume the checkpoint to checkpoint scores are independent and normally distributed (which we observe when computing decision accuracy on intermediate checkpoints in Figure 7). Under this assumption, the ratio between the sample variance and the population variance follows a scaled chi squared distribution:
>
> $\frac{(n-1)s_n^2}{\sigma^2} \sim \chi^2_{n-1}$
>
> Therefore we would like to calculate the probability that the sample standard deviation $s_n$ is within one standard deviation of the population standard deviation $\sigma$:
>
> $|s_n - \sigma| < \sigma$
>
> We can rewrite this inequality:
>
> $\left| \frac{s_n}{\sigma} - 1 \right| < 1 \Rightarrow 0 < \frac{s_n}{\sigma} < 2$
>
> And then, can substitute the chi-squared distribution to compute the likelihood w.r.t. $n$:
>
> $\frac{s_n}{\sigma} \sim \sqrt{\frac{\chi^2_{n-1}}{n-1}}$
>
> $P\left( \sqrt{\frac{\chi^2_{n-1}}{n-1}} < 2 \right)$
>
> $P\left( \chi^2_{n-1} < 4(n-1) \right)$
>
> We can then solve the inequality for the smallest value of $n$ for a particular threshold $\alpha$:
>
> $P\left( \chi^2_{n-1} < 4(n-1) \right)>\alpha$
>
> Solving this inequality numerically with $\alpha=0.95$ for increasing values of $n$, we find that $n=9$ provides the smallest sample size such that the probability that the sample standard deviation (the observed noise) is within one standard deviation of the population standard deviation (the true noise) with 95% confidence.
>
> In addition, we can specify a stricter bound by defining the sample standard deviation to be within $k\cdot \sigma$ of the population standard deviation:
>
> $|s_n - \sigma| < k\cdot \sigma$
>
> We then verify this empirically using our estimate for noise at the 7B scale. If we assume the 30 intermediate checkpoints provide a reasonable estimate of the population standard deviation, we then compute the sample standard deviation $s_n$ for $n < 30$. We re-compute $s_n$ 1000 times for different subsets to calculate the likelihood that the sampled standard deviation is within $k\cdot \sigma$ of the population standard deviation $\sigma$. In the below table, we report this likelihood with tolerances $k\in \{0.2, 1.0\}$ for subsets $n\in \{5, 10, 20\}$ and highlight all results with a likelihood above $0.95$:
>
> | $k$ |       0.2  |        0.2  | 0.2         | 1.0        | 1.0         | 1.0         |
> |:-----------------|-----------:|------------:|:------------|:-----------|:------------|:------------|
> | $n$ (# checkpoints)    |       5    |       10    | 20          | 5          | 10          | 20          |
> | AGI Eval         |       0.42 |        0.61 | **0.95**    | **1.00**   | **1.00**    | **1.00**    |
> | ARC Challenge    |       0.44 |        0.7  | **0.98**    | **1.00**   | **1.00**    | **1.00**    |
> | ARC Easy         |       0.38 |        0.65 | **0.97**    | **1.00**   | **1.00**    | **1.00**    |
> | AutoBencher      |       0.47 |        0.71 | **0.97**    | **1.00**   | **1.00**    | **1.00**    |
> | BBH              |       0.42 |        0.6  | 0.95        | **1.00**   | **1.00**    | **1.00**    |
> | BoolQ            |       0.16 |        0.45 | 0.88        | **1.00**   | **1.00**    | **1.00**    |
> | HumanEval        |       0.52 |        0.79 | **0.99**    | **1.00**   | **1.00**    | **1.00**    |
> | HumanEval+       |       0.47 |        0.76 | **0.99**    | **1.00**   | **1.00**    | **1.00**    |
> | CommonsenseQA    |       0.39 |        0.64 | **0.96**    | **1.00**   | **1.00**    | **1.00**    |
> | DROP             |       0.48 |        0.76 | **0.99**    | **1.00**   | **1.00**    | **1.00**    |
> | GSM8K            |       0.49 |        0.77 | **0.99**    | **1.00**   | **1.00**    | **1.00**    |
> | GSM+             |       0.5  |        0.79 | **0.99**    | **1.00**   | **1.00**    | **1.00**    |
> | GSM Symbolic     |       0.37 |        0.64 | **0.96**    | **1.00**   | **1.00**    | **1.00**    |
> | GSM Symbolic P1  |       0.47 |        0.69 | **0.98**    | **1.00**   | **1.00**    | **1.00**    |
> | GSM Symbolic P2  |       0.32 |        0.57 | 0.94        | **1.00**   | **1.00**    | **1.00**    |
> | HellaSwag        |       0.39 |        0.65 | **0.97**    | **1.00**   | **1.00**    | **1.00**    |
> | Jeopardy         |       0.42 |        0.69 | **0.98**    | **1.00**   | **1.00**    | **1.00**    |
> | MBPP             |       0.43 |        0.63 | **0.96**    | **1.00**   | **1.00**    | **1.00**    |
> | MBPP+            |       0.41 |        0.63 | **0.96**    | **1.00**   | **1.00**    | **1.00**    |
> | MedMCQA          |       0.5  |        0.79 | **0.99**    | **1.00**   | **1.00**    | **1.00**    |
> | Minerva MATH     |       0.38 |        0.53 | 0.93        | **1.00**   | **1.00**    | **1.00**    |
> | Minerva MATH 500 |       0.28 |        0.53 | 0.92        | **1.00**   | **1.00**    | **1.00**    |
> | MMLU             |       0    |        0    | 0.54        | 0.83       | **1.00**    | **1.00**    |
> | MMLU Pro         |       0.51 |        0.78 | **0.99**    | **1.00**   | **1.00**    | **1.00**    |
> | All Tasks        |       0    |        0    | 0.08        | 0.83       | **1.00**    | **1.00**    |
> | Code Tasks       |       0.49 |        0.78 | **0.99**    | **1.00**   | **1.00**    | **1.00**    |
> | Knowledge Tasks  |       0    |        0    | 0.15        | 0.83       | **1.00**    | **1.00**    |
> | Math Tasks       |       0.55 |        0.83 | **0.99**    | **1.00**   | **1.00**    | **1.00**    |
> | OLMES Core 9     |       0.31 |        0.49 | 0.92        | **1.00**   | **1.00**    | **1.00**    |
> | OLMES Gen        |       0.48 |        0.74 | **0.98**    | **1.00**   | **1.00**    | **1.00**    |
> | OpenBookQA       |       0.42 |        0.73 | **0.98**    | **1.00**   | **1.00**    | **1.00**    |
> | PIQA             |       0.43 |        0.69 | **0.98**    | **1.00**   | **1.00**    | **1.00**    |
> | SocialIQA        |       0.3  |        0.44 | 0.88        | **0.99**   | **1.00**    | **1.00**    |
> | SQuAD            |       0.48 |        0.72 | **0.99**    | **1.00**   | **1.00**    | **1.00**    |
> | TriviaQA         |       0.48 |        0.76 | **0.99**    | **1.00**   | **1.00**    | **1.00**    |
> | WinoGrande       |       0.42 |        0.67 | **0.97**    | **1.00**   | **1.00**    | **1.00**    |
>
> In practice, we find that for a large bound ($\pm 1$ std. dev.) can be satisfied for almost all benchmarks with $n=5$ intermediate checkpoints, but for smaller bounds, (20% of $\pm 1$ std. dev.), using $n=20$ gives an adequate estimate for 34 of 39 benchmarks we considered in our work.
>
> For our experiment on the 1B-5xC checkpoints, we estimate noise using the average noise of the last 5 checkpoints for all 25 models, so our estimate of noise considers $5\cdot 25=125$ scores, rather than only 5 scores as the estimate of noise.
>
> We will add these details in our final draft to provide guidance on selecting the minimum number of intermediate checkpoints to estimate the step-to-step noise.
>
> 2. **Clarity of notation**
>
> Thank you for your feedback on our notation, we will review the terms in Section 2 and 3 to ensure clarity. To answer the specific points: the benchmark score $B(\cdot)$ is a scalar score; the subscripts for model $m$ denoted as $m_a$, $m_b$ refer to models trained on two datasets $a$ and $b$; we will use a different subscripts on L160 to denote each pair of models, which is separate from the subscript $i$ used on L146 to denote the intermediate checkpoint.
>
> **Questions:**
>
> > How to choose $n$ in Equation (2)?
>
> Please see our response to concern 1 above. We plan to include this finding in the final draft of our work.
>
> > If a benchmark has high SNR on a set of models $\mathcal{M}$, will it be reliable and useful for decision making on a new model $m$ that is not in $\mathcal{M}$?
>
> The application of SNR to new models outside of the set $\mathcal{M}$ depends on the similarity of $m$ to the models studied. The broadest definition we use is the set of “external models” we study in Appendix C.3. In this set, we calculate SNR using models from different families, but with similar architectures. In this case, we expect the SNR findings to hold to new dense transformer-based models trained to similar compute budgets, but perhaps different data, slight architectural changes (for instance, MLA or GQA) or a different learning rate schedule, as these were varied in the set used to calculate signal.
>
> However, we believe extending this across architectures (e.g., MoEs or state-space models) is a good opportunity for future work.

---

> > ### Comment · Reviewer_Nqe6 · 2025-08-02
> >
> > Thank you for your detailed response! I really enjoy the analysis and experiment in the selection of $n$. I will raise the score to 5.

---

> ### Author Response · Authors · 2025-08-07
>
> Thank you for response and we are glad to learn that we could address most of your points. If you have any outstanding or new questions, we are eager to engage in further discussion.

---

### Official Review · Reviewer_CAzp · 2025-07-02

**Clarity:** 2
**Significance:** 2
**Originality:** 3
**Rating:** 4
**Confidence:** 2

**Summary:**

This paper introduces a framework for analyzing language model evaluation through signal-to-noise ratio (SNR) to reduce uncertainty in benchmark-based decisions. The authors define "signal" as a benchmark's ability to distinguish between models of different quality (measured by relative dispersion of scores) and "noise" as sensitivity to training variability (measured by relative standard deviation across final checkpoints). The paper proposes three interventions: filtering noisy subtasks, averaging checkpoint scores, and using bits-per-byte metrics instead of discrete accuracy measures.

**Questions:**

How sensitive are the results to the specific choice of relative dispersion for measuring signal? Can you test alternative signal metrics that might be less sensitive to outliers?

How well does the SNR framework generalize to other critical decisions in LLM development, such as architecture selection, training procedure optimization, or safety evaluation? (Even theoretical arguments would be nice)

It was a bit unclear to me why relative dispersion and checkpoint-level noise specifically should predict decision accuracy and scaling law reliability?

Can the authors provide a systematic analysis of when these interventions are cost-effective given their computational overhead and the magnitude of improvements achieved?

**Ethical Concerns:**

["NO or VERY MINOR ethics concerns only"]

**Limitations:**

The authors provide a reasonable discussion of limitations, acknowledging the focus on specific experimental settings and the need for future work on other sources of noise.

The evaluation focuses primarily on two specific use cases (dataset selection and scaling law prediction) but doesn't address other critical decision-making scenarios in LLM development, such as architecture choices, training procedure selection, or safety evaluation trade-offs. As well as mentioning about the potential brittleness of the signal metric to outliers and distribution assumptions and theoretical understanding of why these specific definitions work.

**Paper Formatting Concerns:**

The paper is generally well-written

**Quality:**

3

**Strengths And Weaknesses:**

Strengths
The experimental evaluation is comprehensive, encompassing 465 open-weight models from 60M to 32B parameters across 30 diverse benchmarks, resulting in 50K evaluation results. The systematic correlation analysis between SNR and both decision accuracy and scaling law prediction error provides strong empirical validation of the framework's utility.
The three proposed interventions show consistent improvements across multiple evaluation settings. The bits-per-byte intervention is particularly compelling. The checkpoint averaging intervention provides a simple but effective noise reduction strategy.

Weaknesses
The choice of relative dispersion (max difference divided by mean) as the signal metric I believe to be quite problematic. This measure is highly sensitive to outliers and may not accurately capture a benchmark's discriminative power. More robust measures like interquartile range or statistical depth-based metrics would provide better characterization.
Many of the reported improvements, while consistent, are relatively small. For example, the filtering intervention improves MMLU decision accuracy by only 2.6%, and averaging noise improves the 30-task average by 2.4%. Given the computational costs of implementing these interventions, the practical significance may be limited.

---

> ### Author Rebuttal · Authors · 2025-07-31
>
> Thank you for your review and feedback, and for noting the “strong empirical validation of the framework's utility”. We address your feedback and questions below:
>
> **Concerns:**
>
> 1. **Choice of Relative Dispersion**
>
> We would like to emphasize that our work does not argue that one particular metric is the correct way to measure the spread of datapoints, rather, that our contribution is that signal, noise and SNR are useful. When selecting the signal metric, we considered 20 alternatives including some measures that are not sensitive to outliers, with results in Table 2 in Appendix A. Below, we extend these results with interquartile range and its variants, along with 1-dimensional versions of depth-based metrics (Halfspace Depth and Projection Depth).
>
> We calculate the correlation between decision accuracy and SNR calculated using the additional metrics as our measure of signal, and plan to include these additional results in the paper:
>
> | Measure of Signal         |                                             |   SNR $R^2$  |
> |:--------------------------|:--------------------------------------------|-------------:|
> | Rel. Dispersion           | $\max_{i,j} \|c_i - c_j\|/\bar{c}$            |   0.569      |
> | Rel. Std. Dev.            | $\sigma/\mu$                                |   0.566      |
> | Interquartile Range       | $Q_3 - Q_1$                                 |   0.464      |
> | Robust Range              | $P_{95} - P_5$                              |   0.444      |
> | Quartile Deviation        | $(Q_3 - Q_1)/2$                             |   0.453      |
> | Median Absolute Deviation | $\text{median}(\|c_i - \text{median}(c)\|)$   |   0.417      |
> | Halfspace Depth           | $\min\left( F_n(x),\ 1 - F_n(x) \right)$ where $F_n(x) = \frac{1}{n} \sum_{i=1}^n \mathbb{I}[c_i \leq x]$ |   0.036  |
> | Projection Depth          | $\left( 1 + \frac{\|x - \text{med}(c)\|}{\text{MAD}(c)} \right)^{-1}$                                       |   0.033  |
>
> For measures that are not sensitive to outliers such as quartile deviation $(Q_3 - Q_1)/2$ and median absolute deviation $\text{median}(|c_i - \text{median}(c)|)$, we found that using these as measures of signal resulted in a slightly lower, but similar, correlation with decision accuracy as relative dispersion ($R^2 = 0.4528$ and $R^2 = 0.4168$ respectively).
>
> Our data in Appendix A.2.1 and the table above show that different metrics lead to similar conclusions, so we argue our work is not particularly sensitive to the choice of metric. Finally, we would like to reiterate that our contribution is the framework itself to capture these complementary notions of signal and noise, and that our framework is general enough to support plug-and-play with different signal metrics depending on the use case.
>
> 2. **Magnitude of Improvement**
>
> The +2.6% improvement of decision accuracy on MMLU from 91.5% to 94.1% shows that SNR can further improve the sensitivity of a widely used benchmark with an already high decision accuracy, which is a notable 31% error reduction, $0.31=\frac{(1-0.915)-(1-0.941)}{(1-0.915)}$. We believe reducing this likelihood of reporting a false ranking of models at a small scale is useful, and demonstrates the utility of SNR.
>
> Additionally, we find that SNR is a powerful tool for detecting poorly performing task setups along with improving already high quality benchmarks. For example, we show the decision accuracy on the 30-task average improves from 77% to 83.7% (+6.7%) using the bits-per-byte task format. For individual tasks like Minerva MATH, we show up to +49% improvements in decision accuracy (51% to 90%).
>
> 3. **Computational Cost**
>
> First, we remark that calculating SNR is a one-time cost, not something that needs to be recalculated with every new model developed. To clarify, we do not propose SNR as a way to improve model performance on a given benchmark. Rather, ML practitioners building new models can simply use the conclusions from previously-calculated SNR values.
>
> For example, if a benchmark has high SNR for a set of 1B models, a practitioner developing a new 1B model could reasonably decide to use that benchmark and compare against other 1B models. On the other hand, if a benchmark has low SNR, the ML practitioner could choose not to evaluate on it. A researcher might also examine a benchmark that has low SNR to find some problem that they can fix (e.g., incorrect annotations or otherwise problematic data); once any problems are fixed and the benchmark has higher SNR, that fixed benchmark can be used in perpetuity without incurring any additional cost.
>
> Compared to other methods of measuring the quality of a benchmark, we believe SNR requires far less computation. For example, performing a DataDecide-style experiment at the 7B scale would require training multiple 7B models on different training datasets, which is too expensive at that scale to be a practical method for measuring evaluation quality. Instead, one can compute SNR only for the evaluation cost of the checkpoints used for signal and noise. In practice, our cost for computing SNR at the 7B scale (42 models) on the full 30-task evaluation suite required 789 NVIDIA H100 hours. Compared to the development costs for other 7B models such as OLMo 2, which reported 269K H100 hours in total [1], our SNR analysis on all benchmarks requires 0.29% of the total development cost. Given the enormous costs of model training, we believe the compute cost associated with improving evaluation procedure is well worth the improvements we show in our work.
>
> Additionally, in Figure 6 (Appendix B) we show that other estimates of modeling noise, such as seed noise and data order noise, highly correlate with the step-to-step noise of only the final checkpoints. This one-time cost allows us to further reduce the cost as one can get an estimate of noise without training models across random seeds.
>
> While the evaluation cost is non-trivial, it allows estimating the quality of benchmarks at much larger scales without training a population of models from-scratch.
>
> [1] Holistically Evaluating the Environmental Impact of Creating Language Models (Morrison et al., ICLR 2025)
>
> **Questions:**
>
> > How sensitive are the results to the specific choice of relative dispersion for measuring signal? Can you test alternative signal metrics that might be less sensitive to outliers?
>
> Please see our response to concern 1 above. We extensively evaluated 20 different metrics for our signal term, which we discuss in Appendix A, and include additional results on alternative signal metrics in the above table.
>
> > How well does the SNR framework generalize to other critical decisions in LLM development, such as architecture selection, training procedure optimization, or safety evaluation?
>
> We extend SNR to a large set of models in Table 3 as part of Appendix C.3, with the results on external models across compute scales up to $10^{24}$ estimated FLOPs in Figure 11. In this setting, we use results across model families trained with similar compute budgets. This provides an estimate of SNR across different training procedures, although we believe explicitly extending this across architectures (e.g., MoEs or state-space models) is a good opportunity for future work.
>
> However, our work primarily uses SNR in the context of evaluating data decisions, as this family of decisions is particularly sensitive to downstream evaluation. This follows existing work that shows that held-out language modeling loss is a good predictor of downstream performance for architecture and training decisions, but that the mapping from validation loss to downstream performance varies between training datasets [1]. Therefore, we anticipate on these modeling and architectural decisions, there is less need to rely on downstream evaluation for decision-making compared to validation loss.
>
> [1] LLMs on the Line: Data Determines Loss-to-Loss Scaling Laws (Mayilvahanan et al., ICML 2025)
>
> > … why relative dispersion and checkpoint-level noise specifically should predict decision accuracy and scaling law reliability?
>
> To clarify our contribution, our framework proposes that, in general, some measures of signal and noise are predictive of decision accuracy and scaling law error, rather than only the two particular measures we use to quantify signal and noise (the relative dispersion and checkpoint-level noise).
>
> In Table 2, we show that many different measures of signal exhibit similar correlation with decision accuracy when used as the signal term in SNR. And in Figure 6, we find that different sources of noise (e.g., the seed noise, data order noise, and checkpoint-to-checkpoint noise) all have high correlation. So, many different measures could be used to capture signal and noise.
>
> > Can the authors provide a systematic analysis of when these interventions are cost-effective given their computational overhead and the magnitude of improvements achieved?
>
> Please see our response to concern 2, on improvement, and concern 3, on computation cost. Additionally, we emphasize that calculating SNR is a one-time cost to evaluate the quality of a benchmark. For instance, when a researcher is developing a new benchmark, they likely want to evaluate a suite of models to showcase that their benchmark is useful. In this case, calculating SNR only requires the cost of the evaluation, and can indicate how well the benchmark can distinguish between model decisions. The researcher might find that small (e.g., 1B) models evaluated on their benchmark have low SNR, while large (e.g., 32B) models have high SNR, in which case they could describe their new benchmark as useful for evaluating models trained with large compute budgets. On the other hand, a researcher might find low SNR across model scales, which may indicate a need to investigate annotation errors, formatting or other problems in their benchmark.

---

> ### Author Response · Authors · 2025-08-07
>
> Thank you again for reviewing our work. We would be eager to hear whether we have addressed your outstanding questions on our work, and if you have any additional questions.

---

### Official Review · Reviewer_bZgC · 2025-07-03

**Clarity:** 4
**Significance:** 3
**Originality:** 3
**Rating:** 5
**Confidence:** 3

**Summary:**

The paper analyzes specific properties which make a benchmark more reliable and useful for assessing LLMs, and interventions to design higher-quality evaluation benchmarks.
They discuss two key metrics of a benchmark : signal and noise. Signal measures the ability of the benchmark to differentiate between better and worse models, while noise measures the variability of different training steps of a given model on the benchmark. They show that benchmarks with high SNR are more reliable to extrapolate decisions made at smaller models to larger models, and less noise leads to lower scaling law prediction error. To use this insight to design better benchmarks, they try some interventions like using metrics which have higher SNR, and averaging the model outputs from multiple checkpoints. They also identify smaller subsets of existing benchmarks, which have a higher SNR, and improve the performance on both the tasks, despite having less than half the number of samples of the full benchmark

**Questions:**

- By definition, computing the “signal” of a benchmark assumes that the models being evaluated actually differ in their quality/abilities. How does one determine if this is truly the case for a given set of models? It seems like a recursive problem, to select the right set of models to evaluate the benchmarks, and the right benchmarks to evaluate the models?

- For the experiments on averaging checkpoint-to-checkpoint noise, the text in section 5.2 says that you average the target model in both settings, and only change whether you avg the smaller models used to fit the scaling law. However, in Table 1, right table, you use the term “Avg Train”, which is not clear. Does it mean averaging both the small and target models?

**Ethical Concerns:**

["NO or VERY MINOR ethics concerns only"]

**Final Justification:**

I stand by my initial rating of 5 (Accept), after considering the discussions with the authors in the rebuttal phase.

**Limitations:**

Yes

**Quality:**

3

**Strengths And Weaknesses:**

Strengths

- The problem discussed in this paper is highly relevant in current time, when new LLMs and benchmarks are released at a rapid rate, making it hard to determine which benchmarks are reliable for comparing different LLMs. It is even more useful for someone developing LLMs : training smaller language models and evaluating them on benchmarks to extrapolate the performance of the bigger models is a very common practice, since it’s infeasible to experiment with training the full model directly.
- There are extensive investigations done to tease out the effect of signal and noise on the decision accuracy and scaling law prediction error. The figures related to the investigations are comprehensive.
- The paper is very well-written, with clear descriptions of the key terms like signal, noise, decision accuracy, scaling law prediction error. I like that the authors explored different notions of noise, instead of just choosing one.
- The hypotheses are well investigated through extensive experiments.

---

> ### Author Rebuttal · Authors · 2025-07-31
>
> Thank you for your review and noting that “[t]he paper is very well-written” and that “[t]he hypotheses are well investigated through extensive experiments”! We address your questions below:
>
> **Questions:**
>
> > By definition, computing the “signal” of a benchmark assumes that the models being evaluated actually differ in their quality/abilities. How does one determine if this is truly the case for a given set of models?
>
> This is a good question, and gets at an important idea: specific measurements of signal and noise are really only valid for the set of models and benchmarks used to make those measurements. They might not generalize to other models, especially if those other models are significantly different than the ones used to calculate signal and noise (e.g., they are significantly larger or smaller, or have significantly different capabilities). That said, SNR can be used to find generalizable claims; for example, if a set of models reasonably should be able to do a given task but have low SNR on a benchmark, a researcher might find problems with the format or annotations on the benchmark which they could then fix (and could have been problematic for any model evaluated on that benchmark). We do not need to assume that models differ in a particular skill to compute the signal of a benchmark. We only need to assume these models are drawn from a population of models one wants to study.
>
> For example, one may want to compute SNR for a benchmark using models trained on  a set of data recipes at the 1B scale. Assume these particular recipes only vary sources of web data, and do not contain code. Therefore, the resulting models may show very similar scores on HumanEval, near random chance. This may not indicate that HumanEval is a problematic benchmark, but that it cannot distinguish between those particular models.
>
> When we compute SNR at larger scales (Table 3), the external models we used were selected with the broad assumption that the models may have been trained with different data, architecture, learning rate schedules, etc. This allows benchmarks to have a higher spread if scores are sensitive to any of those modeling decisions.
>
> > Does [“Avg Train” in Table 1, right] mean averaging both the small and target models?
>
> That is correct, we average the final checkpoints of the target model for the scaling law experiments in Section 5.2. In Table 1, right, both columns average the “train” models (the models used to fit the scaling law prediction). We will explicitly note this in the writing.

---

> > ### Comment · Reviewer_bZgC · 2025-08-07
> >
> > Thanks for the clarifications! I am satisfied with the answers provided by the authors.

---

> ### Author Response · Authors · 2025-08-07
>
> Thank you again for reviewing our work. We would be eager to hear whether we have addressed your outstanding questions on our work, and if you have any additional questions.

---

### Official Review · Reviewer_RGQH · 2025-07-04

**Clarity:** 3
**Significance:** 4
**Originality:** 2
**Rating:** 5
**Confidence:** 4

**Summary:**

This work tackles a “meta-evaluation” problem: how to evaluate the reliability of benchmark suites when we must make model-selection or scaling decisions from small-scale experiments. The authors design two complementary properties of a benchmark—signal (its power to separate stronger from weaker models) and noise (its sensitivity to stochastic variation such as checkpoint choice)—and show that the signal-to-noise ratio (SNR) strongly predicts decision accuracy and scaling-law extrapolation error across 30 diverse benchmarks and 465 open-weight language models (60 M – 32 B parameters). They then introduce four practical interventions—metric replacement (e.g. perplexity over accuracy), noisy-benchmark filtering, checkpoint averaging, and other dataset refinements—that each raise SNR and yield more robust conclusions.

**Questions:**

Could you provide examples of the datapoints you filtered out (e.g. from MBPP) and clarify whether they were incorrect, ambiguous, or otherwise problematic?

**Ethical Concerns:**

["NO or VERY MINOR ethics concerns only"]

**Final Justification:**

Thank you for the detailed response. It clarifies the contribution of the paper compared to prior work and EvalArena, and addressed some of my concerns, including those related to spearman Correlation. I also appreciate the case study and I like the point of using the SNR to identify potentially problematic test cases in benchmarks. Therefore, I raised my score.

**Quality:**

2

**Strengths And Weaknesses:**

Strengths:

1. Rather than analyzing individual models or tasks, the paper tackles a "meta-evaluation" problem: it studies how we reason from small models to large ones using an evaluation benchmark, a rarely explored but important area.
2. Using pairwise ranking accuracy to quantify decision reliability avoids strong distributional assumptions and is easy to interpret.
3. Section 3.1 convincingly shows that averaging the final n checkpoints is a stable and unbiased proxy for intrinsic variance under some assumptions (same/similar learning rate schedules).

Weaknesses:

1. The Decision Accuracy experiment is very close to the DataDecide paper, and the Scaling-Law Prediction Error setup mirrors prior work. More discussion/comparison is required to acknowledge the contributions in prior work and discuss novel contributions in this paper.
2. The custom “Decision Accuracy” formula feels ad-hoc and non-standard; a rank-based statistic such as Spearman’s Correlation Coefficient (https://en.wikipedia.org/wiki/Spearman%27s_rank_correlation_coefficient) is similar, and might be more familiar.
3. Missing comparison to EvalArena. EvalArena (https://github.com/crux-eval/eval-arena) also analyzes benchmark signal-to-noise ratio but is not cited or compared.

---

> ### Author Rebuttal · Authors · 2025-07-31
>
> Thank you for your review and noting that our work addresses “a rarely explored but important area”! We address your feedback and questions below:
>
> **Concerns:**
>
> 1. **Novelty w.r.t. Decision Accuracy and Scaling Law Error**
>
> Our core contribution is the signal to noise (SNR) framework, and the interventions to improve SNR. You are correct that decision accuracy and scaling law prediction error are from previous work, and this is intentional as our goal is to show that SNR is useful for language model development practices in existing literature. Our work does not compete with or aim to replace previous work on scaling laws or decision-making processes, instead we aim to build on that work by introducing a tool, SNR, which can be used to understand which benchmarks are appropriate (or not) for scaling laws or approaches like decision accuracy.
>
> Moreover, scaling laws and decision accuracy are two instantiations of more general steps that play a key role in building ML models. For example, scaling laws are an example of an attempt to directly estimate model performance, while decision accuracy is an example of ranking training data candidates. We believe our contribution, SNR, captures information that would be useful to other approaches beyond these specific metrics, though we leave that to future work. In our final draft, we will clarify that Sec. 2.1 is presenting background work prior to our discussing our core method in Sec. 3. Additionally, we will give more discussion at the end of Sec. 1 to further highlight our contributions mentioned in the abstract and discussed later in the paper.
>
> 2. **Choice of Decision Accuracy over other Rank Statistics**
>
> We use the decision accuracy metric for two reasons: first, to mirror its use in the DataDecide paper, and second because a particular value for decision accuracy is more interpretable than a correlation coefficient. For example, decision accuracy of 80% means that 80% of pairwise comparisons between different data mixes correctly predict which is better at large scale, but 0.8 correlation doesn’t have a similarly grounded interpretation. That said, decision accuracy is a rank-based statistic, in fact it is proportional to Kendall’s Tau modulo a scale and shift: $\tau = 2 \cdot (\text{decision accuracy}) - 1$, which we briefly show below and will add to the final revision of our work:
>
> Kendall’s Tau is defined using the number of concordant pairs $C$ and discordant pairs $D$ between two rankings.
>
> $$\tau = \frac{C - D}{\binom{N}{2}}$$
>
> Decision accuracy is defined as only the number of concordant pairs $C$:
>
> $$\text{decision accuracy} = \frac{C}{\binom{N}{2}}$$
>
> Since we do not allow ties, $C$ and $D$ make up the total number of pairs $\binom{N}{2} = C + D$, we can rewrite decision accuracy as follows:
>
> $$\tau = \frac{C - (\binom{N}{2} - C)}{\binom{N}{2}} = \frac{2C - \binom{N}{2}}{\binom{N}{2}} = 2 \cdot \frac{C}{\binom{N}{2}} - 1$$
> $$\tau = 2 \cdot (\text{decision accuracy}) - 1$$
>
> Additionally, we reproduce the experiment in Figure 2, which calculates the correlation with SNR and the decision accuracy, by using SNR to predict the Spearman rank correlation of the 25 DataDecide mixes instead:
>
> | Signal Measure   |   SNR R² (Spearman) |   SNR R² (Kendall) |   SNR R² (Decision Acc) |
> |:-----------------|--------------------:|-------------------:|------------------------:|
> | Rel. Dispersion  |              0.4902 |             0.5687 |                  0.5687 |
>
> We find the $R^2$ value to show that SNR is predictive of both the original Decision Accuracy setting and when using the Spearman correlation. We will include this in the final revision of our work.
>
> As our central contribution was a framework to improve existing methods of evaluating data decisions, we did not change the existing rank statistic used in DataDecide. However, we agree that other rank-correlation statistics could be used to evaluate the interventions we propose for improving signal-to-noise.
>
>
> 3. **Comparison to EvalArena**
>
> Thank you for highlighting this work! We will include a discussion in our final draft.
>
> While EvalArea defines a term “signal-to-noise ratio”, their definition is significantly different from ours. Their term is calculated using the scores of final checkpoints for two models within the same model family (e.g., 7B and 70B variants of the same model). Our work instead defines noise as the checkpoint-to-checkpoint variance of one model evaluated on an entire benchmark during training and signal as the spread of a population of models trained at the same compute budget.
>
> Conceptually, our measure uses modeling noise, in the form of checkpoint-to-checkpoint noise, while EvalArea’s measure of noise is based on the statistical variability of the dataset. A benchmark may have low statistical variability but high checkpoint-to-checkpoint noise (e.g., a benchmark with large sample size but also a high inter-instance correlation, which we observe with BoolQ). Also, our measure of SNR is calculated at a particular model scale (1B, 7B, 13B, etc.), instead of using the difference of scores between model scales to assign a single score to a benchmark. This leads to distinct values for SNR at each model size, as we find in Table 3, which allows us to state which benchmarks are useful at a particular scale.
>
> Empirically, we can briefly validate that the EvalArea formulation of SNR will result in a different ranking of benchmarks by using the experiment in Figure 2 (right). In this setting, we calculate signal-to-noise using the implementation in the EvalArea setup for each small DataDecide model {60M, 90M, 150M, 300M, 530M, 750M} to the 1B model. We then calculate the correlation between the SNR term proposed in our work, and discussed in EvalArena, with decision accuracy:
>
> | Measure of SNR                                 | $R^2$ with Decision Accuracy at 1B |
> |:-----------------------------------------------|----------------------------------:|
> | Signal-to-Noise in our work                    | 0.569                             |
> | EvalArena Signal-to-Noise                      | 0.181                             |
>
> While we believe the statistical measures discussed in EvalArena are important indicators of dataset noise, we find in the above table that our SNR approach, based on measuring the checkpoint-to-checkpoint modeling noise, is more predictive in the decision accuracy setting.
>
> Finally, our work goes beyond measuring the noise of existing benchmarks by applying our measure of SNR by designing interventions to improve the evaluation procedure (in Section 5). We believe EvalArena’s work and other related work discussed in our Sec. 6, such as estimating dataset noise using bootstrap confidence intervals, are complimentary methods to the signal-to-noise framework we propose in our work.
>
> **Questions:**
>
> > Could you provide examples of the datapoints you filtered out (e.g. from MBPP) and clarify whether they were incorrect, ambiguous, or otherwise problematic?
>
> Thank you for this question. We will include examples of instances in our final draft. We note that our filtering experiment in Section 5.1 excludes subtasks of MMLU and AutoBencher rather than individual instances; to address this question more fully in the paper we will include the following analysis.
>
> In Figure 15 (Appendix D), we include the exact subtasks filtered in the experiment ranked by their signal-to-noise ratio, which correspond to the subsets that were included in Figure 4.
>
> We can use prior work that has explored the quality of MMLU subtasks to indicate whether the filtered subtasks may have been problematic. If we compare the list of 57 MMLU subtasks ranked by SNR (Figure 15, top left) to the 57 MMLU subtasks listed by the number of errors annotated as part of MMLU Redux (Table 5 of [1]), we find that out of the 20 MMLU subtasks which contain errors in least 5% of instances, 10/20 of these subtasks are also in the lowest 20 tasks sorted by their signal-to-noise ratio. This presents evidence that low SNR may indicate low quality tasks, and we believe this is a good opportunity for future work in evaluation development.
>
> Complimentary to the experiment in Sec. 5.1, we believe that SNR can be helpful for identifying problematic evaluation setups beyond filtering subtasks. For example, in Figure 8 in Appendix C.1 we find that BoolQ has a particularly low SNR despite having a large sample size. We believe this is due to the benchmark formatting only including ‘True’ and ‘False’ continuations where small models can perform particularly well by only guessing the majority class.
>
> [1] Are We Done with MMLU? (Gema et al., NAACL 2025)

---

> ### Author Response · Authors · 2025-08-07
>
> Thank you for acknowledging our rebuttal. If you have any outstanding or new questions, we are eager to engage in further discussion.

---

### Note · Authors · 2025-08-16

We would like to thank the reviewers again for their feedback, and would like to summarize the the discussion:

All reviewers highlighted the strengths of our work: Reviewer RGQH wrote that ‘the paper tackles a "meta-evaluation" problem … a rarely explored but important area.’ Reviewer bZgC noted that “[t]he paper is very well-written,” and Reviewer CAzp highlighted the “strong empirical validation of the framework's utility.” Finally, reviewer Nqe6 noted the “abundant experiments” and that “the results are convincing.” We believe our response addressed the concerns and questions raised by all reviewers.

Our response to Reviewer RGQH added a comparison to additional rank agreement metrics, and a comparison of our method to EvalArea. We also presented additional measures of signal and comparisons, and will include in our final work. For Reviewer CAzp, we clarified the improvements introduced in our work, and the computational cost of our method. Finally, in our response to Reviewer Nqe6, we presented theoretical and empirical guidance on selecting the hyper-parameter $n$ in the noise term, which led to Reviewer Nqe6 increasing their score.

---

### Decision · Program_Chairs · 2025-09-17

**Decision:**

Accept (spotlight)

**Comment:**

This paper introduces a framework for evaluating benchmarks through the lens of signal (a benchmark’s ability to separate stronger from weaker models) and noise (its sensitivity to random training variability). The authors thus propose the signal-to-noise ratio (SNR) as a predictor of benchmark reliability. With a wide range of benchmarks and open-weight models, the authors demonstrate that SNR correlates strongly with decision accuracy and scaling law prediction error, and propose practical interventions.

This paper is quite timely, as (fair) LLM evaluation is a pressing issue. Multiple reviewers appreciated its novelty as a “meta-evaluation” study and its practical utility for practitioners. Weaknesses noted include limited novelty in some experimental setups (which mirror prior work), sensitivity of the chosen “signal” metric to outliers, modest improvements in some settings, and some clarity issues in definitions and notation. During the rebuttal, the authors clarified their contributions relative to prior work, and provided both theoretical and empirical explanations, which successfully addressed reviewer concerns.

Overall this paper makes a technically solid and impactful contribution to the methodology of benchmark design and evaluation. Considering the reviewers' comments and the timely topic, I recommend accept with spotlight.